



# The important roles of surface tension and growth rate in the contribution of new particle formation (NPF) to cloud condensation nuclei (CCN) number concentration: evidence from field measurements in southern China

Mingfu Cai[1,2,3,4], Baoling Liang[3], Qibin Sun[3], Li Liu[4], Bin Yuan[1,2*], Min Shao[1,2], Shan Huang[1,2], Yuwen Peng[1,2], Zelong Wang[1,2], Haobo Tan[4], Fei Li[4,6], Hanbin Xu[3], and Jun Zhao[3,5,7*]

[1] Institute for Environmental and Climate Research, Jinan University, Guangzhou, Guangdong 511443, China

[2] Guangdong-Hongkong-Macau Joint Laboratory of Collaborative Innovation for Environmental Quality, Guangzhou, Guangdong 511443, China

[3] School of Atmospheric Sciences, Guangdong Province Key Laboratory for Climate Change and Natural Disaster Studies, and Institute of Earth Climate and Environment System, Sun Yat-sen University, Zhuhai, Guangdong 519082, China

[4] Institute of Tropical and Marine Meteorology/Guangdong Provincial Key Laboratory of Regional Numerical Weather Prediction, CMA, Guangzhou 510640, China

[5] Southern Marine Science and Engineering Guangdong Laboratory (Zhuhai), Zhuhai, Guangdong 519082, China

[6] Laboratory of straits meteorology, Xiamen, Guangdong 361012, China

[7] Guangdong Provincial Observation and Research Station for Climate Environment and Air Quality Change in the Pearl River Estuary, Guangzhou, Guangdong 510275, China

*Corresponding authors: Bin Yuan (byuan@jnu.edu.cn) and Jun Zhao (zhaojun23@mail.sysu.edu.cn)



**Abstract.**
The contribution of new particle formation (NPF) to cloud condensation nuclei (CCN) number
concentration varies largely under different environments, depending on several key factors such as
formation rate (J), growth rate (GR), distribution of preexisting particles and properties of new particles
during NPF events. This study investigated the contribution of NPF to the $N_{CCN}$ and its controlling factors
based on measurements conducted at the Heshan supersite, in the Pearl River Delta (PRD) region of
China during fall-time 2019. The size-resolved cloud condensation nuclei activity and size-resolved
particle hygroscopicity were measured by a cloud condensation nuclei counter (CCNc) and a hygroscopic
tandem differential mobility analyzer (HTDMA), respectively, along with a scanning mobility particle
sizer (SMPS) and a diethylene glycol scanning mobility particle sizer (DEG-SMPS) for particle number
size distribution (PNSD). A typical NPF event on 29[th] October was chosen to investigate the contribution
of the NPF to $N_{CCN}$ under several supersaturation ratios. Two particle properties (hygroscopicity and
surface tension) affect CCN activation with the latter being more important in terms of the CCN
concentration ($N_{CCN}$). A lower value of surface tension (i.e., 0.06 N m$^{-1}$) than pure water assumption
(0.073 N m$^{-1}$) could increase the $N_{CCN}$ at SS=1.0% by about 20% during non-event period and by about
40% during the event. In addition, an earlier peak time corresponding to a lower critical diameter ($D_{50}$)
was also observed. The results show that high formation rate, growth rate, and low background particle
concentration lead to high number concentrations of newly-formed particles. The high growth rate was
found to have the most profound impact on the $N_{CCN}$ which can be attributed to the facts that a higher
growth rate can grow particles to the CCN size in a shorter time before they are scavenged by pre-existing
particles. Two other NPF events (an event on 18[th] October in this campaign and an event on 12[th]
December, 2014 in Panyu) were chosen to perform sensitivity tests under different scenarios (growth



rate, formation rate, and background particle concertation). The calculated $N_{CCN}$ at SS=1.0% on 12th
December, 2014 was significantly lower than that from the other two events. The event on 12th December
was re-simulated using high growth rate taken from the event on 18th October which resulted in similar
CCN concentrations between the two events (12th December and 18th October), implying that the growth
rate is the most controlling factor for CCN activation. Our results highlight the importance of growth rate
and surface tension when evaluating the contribution of NPF to the $N_{CCN}$.

## 1 Introduction

Atmospheric particles have direct effects on global climate by adsorbing and scattering solar

radiation, while they can act as cloud condensation nuclei (CCN) and exert influences on cloud formation,
life cycle, and albedo, hence indirectly affect the global radiation balance. In general, atmospheric
particles have a cooling effect on the global climate, although the highest uncertainty exists on their
climatic contribution among all the climatic forcings (Stocker et al., 2013). The relationship between the
CCN number concentration ($N_{CCN}$) and its climatic effect represents one of the major uncertainties and
challenges in evaluating the aerosol climatic effect. It is hence essential to carry out field measurements
to investigate the CCN activity and its controlling factors.

New particle formation (NPF) as an important source of global atmospheric particles, is frequently

observed in various atmospheric environments, including forest, urban, and agricultural regions
(Kulmala et al., 2004). Once formed, the particles can grow to the CCN sizes (50-100 nm) within a few
hours and contribute significantly to the $N_{CCN}$ (Leng et al., 2014; Spracklen et al., 2008; Dameto de
España et al., 2017). The extent to which newly-formed particles can contribute to the $N_{CCN}$ is controlled
by many factors, including formation rate (J), growth rate (GR), background particle number size



distribution (PNSD), and properties of the particles. The formation rate is defined as a flux of newly-
formed particles at a threshold diameter and is usually used to describe how many particles are produced
into the atmosphere during an event. The growth rate (GR) represents the diameter change of the particles
in a certain time period, and particles with a higher GR will grow to the CCN sizes in a shorter time. The
background PNSD controls the scavenging of the newly-formed particles, and the high concentration of
pre-existing particles will efficiently scavenge these particles before they can grow to the CCN sizes.
The properties of the particles (e.g., chemical composition, hygroscopicity, and surface tension) affect
their ability of acting as CCN. In general, particles containing a higher fraction of inorganic matters or
water-soluble organics are more hygroscopic and are more easily activated due to a lower critical
diameter ($D_{50}$). Recent studies showed that surfactant effects of organic matters were found on the
particle surface which could lead to an increase of the CCN activity (Ovadnevaite et al., 2017; Cai et al.,
2018; Liu et al., 2018). The contribution of NPF to $N_{CCN}$ is difficult to be quantitatively evaluated and
currently the controlling factors are not fully understood, constraining an accurate quantification of the
aerosol climatic forcing from NPF.

NPF event was well known to have an important contribution to the $N_{CCN}$, while a wide range of

$N_{CCN}$ during NPF events was reported in the literature. Yue et al. (2011) showed that the $N_{CCN}$ during
NPF events was increased by a factor of 0.4-6 in Beijing. However, much less (a factor of 1.17-1.88)
increase of the $N_{CCN}$ was observed during NPF events in Shanghai (Leng et al., 2014). The results from
Ma et al. (2016) showed that the $N_{CCN}$ was significantly impacted by the hygroscopicity of newly-formed
particles during NPF events in the North China Plain (NCP). Yu et al. (2014) reported an average factor
of 4.7 increase of the $N_{CCN}$ during NPF events from growth of new particles to the CCN sizes in Ozark
forest. Rose et al. (2016) showed that NPF could be a larger contributor to $N_{CCN}$ compared to transport



in free troposphere. A long-term field measurement in the urban Vienna conducted by Dameto de España
et al. (2017) reported that the $N_{CCN}$ (at 0.5% ss) could increase up to 143% during NPF events.
Kalkavouras er al. (2017) found that the NPF could double the $N_{CCN}$ (at 0.1% ss), but could augment the
potential droplet number only by 12%.
Factors that control the CCN activity of newly-formed particles (formation rate, growth rate, and
particle properties) were investigated worldwide. These parameters varied substantially in a large
temporal and spatial scale. For example, the mean formation rate of 10 nm particles ($J_{10}$) was 3.7 cm$^{-3}$
s$^{-1}$ in Nanjing (An et al., 2015), which was much higher than that (0.40 cm$^{-3}$ s$^{-1}$) reported in Shanghai
(Leng et al., 2014). A value of 3.3~81.4 cm$^{-3}$ s$^{-1}$ was reported for the mean formation rate of 3 nm particles
($J_3$) based on one-year long measurements in Beijing (Wu et al., 2007). In the NCP region, a long-term
measurement conducted by Shen et al. (2011) reported that the $J_3$ ranged from 0.7 to 72.7 cm$^{-3}$ s$^{-1}$, with
a mean value of 8.0 cm$^{-3}$ s$^{-1}$ . Shen et al. (2019) reported an average $J_3$ value of 1.30 cm$^{-3}$ s$^{-1}$ at Mountain
Tai, which was much lower than urban regions. The growth of newly-formed particles can be
characterized by the particle growth rate. Kulmala et al. (2004) summarized a wide range of growth rate
(1 to 20 nm h$^{-1}$) from more than 100 filed measurements of NPF in mid-latitudes. On the one hand, the
growth rates are usually high in polluted region, for example, a growth rate of 11.6-18.1 nm h$^{-1}$ was
reported in New Delhi, India (Kulmala et al., 2005;Mönkkönen et al., 2005). On the other hand, the
growth rates are in general low in forest regions, for example, a median value of 2.5 nm h$^{-1}$ was reported
from long term measurements (Nieminen et al., 2014). Furthermore, large uncertainties exist for the
measured growth rates even in the same region. For example, the growth rates under sulfur-poor
conditions were about 80% higher than those under sulfur-rich conditions in Beijing (Yue et al., 2011).
The condensable vapors not only control the growth rate, but also decide the hygroscopicity of newly-



formed particle, which can vary in a large range from event to event. Wu et al. (2013a) reported a
hygroscopic growth factor of 1.2 to 1.55 during NPF events in a mountain region, Germany. Asmi et al.
(2010) found a significant contribution of organic vapors to particle formation and growth, leading to a
low hygroscopicity of newly-formed particles in the Antarctica region. The above studies show large
temporal and spatial variations of characteristics in the properties of newly-formed particles (i.e., the
formation rate, growth rate and hygroscopicity) during NPF events. However, how these parameters
contribute to the variation of the $N_{CCN}$ during NPF events in various regions is yet to be investigated.
Although the Pearl River Delta region (PRD), one of the most economically developed areas in
China, has made substantial progress in mitigating haze pollution, especially in achieving $PM_{2.5}$ national
level II standard (an average annual mass concentration of less than 35 μg m$^{-3}$ for $PM_{2.5}$), the intensive
human activities and photochemistry lead to emissions and productions of a large amount of condensable
air pollutants for initiating formation of atmospheric particles and promoting their subsequent growth.
Several studies reported the frequent occurrences of NPF events in urban and rural areas of the PRD
which provide a large amount of particles to the local atmosphere (Yue et al., 2013;Liu et al., 2008;Yue
et al., 2016;Wang et al., 2013). However, these studies focused primarily on the characteristics of the
NPF events, the contribution to the $N_{CCN}$ and the controlling factors were still unknown, hindering an
accurate assessment of NPF in CCN formation and eventually global climate change.
In this study, we analyzed the contribution of NPF to the $N_{CCN}$ based on a rural field campaign
conducted at the Heshan supersite in the PRD region during Fall (October and November, 2019). A suite
of advanced analytical instruments were employed to measure particle hygroscopicity, size-resolved
CCN activity, and particle number size distribution (1 nm - 10 μm). Here, we select three representative
NPF events (two from this measurement, the other one from a previous measurement in Panyu,



Guangzhou, 2014) to quantitively investigate the contribution of NPF to the $N_{CCN}$ and impact factors (i.e.,
formation rate, growth rate, background particle concentration, and particle properties) that manipulate
the contribution.

## 2    Measurement site, instrumentation, and methodology

### 2.1  Measurement site

The field campaign was conducted at the Heshan supersite in the Guangdong Province of China

during the Fall season (from 27$^{th}$ September to 17$^{th}$ November, 2019). This rural site (22°42′39. 1″N,
112°55′35.9″E) is located at the southwest of the PRD region (about 70 km away from megacity
Guangzhou) with an altitude of about 40 m above sea-level and the site is surrounded by several farms
and villages. All the instruments were placed in an air-conditioned room (T=298K) on the top floor of
the building at the supersite, administrated by Guangdong Provincial Environmental Monitoring Centre.
An aerosol sampling port equipped with a PM$_{10}$ cyclone inlet was made of a 6 m long 3/8″ o.d. stainless-
steel tube. The sampling air was dried to a relative humidity (RH) lower than 30% by passing through a
Nafion dryer (model MD-700, Perma Pure, Inc., USA) before the air entered into the individual
instruments.

### 2.2 Instrumentation

### 2.2.1 Particle number size distribution and size-resolved CCN activity measurements

The particle number size distribution (PNSD) in a complete size range of 1 nm- 10 μm (an upper

cut size of 10 μm) was measured by a diethylene glycol scanning mobility particle sizer (DEG-SMPS,



153 model 3938E77, TSI Inc., USA), a SMPS (model 3938L75, TSI Inc., USA), and an aerodynamic particle

154 sizer (APS, model 3321, TSI Inc., USA). The DEG-SMPS was applied to measure particles with a size

155 range of 1-30 nm, consisted of a nano-differential mobility analyzer (nDMA, model 3086, TSI Inc., USA),

156 a nano enhancer (model 3777, TSI Inc., USA), and a condensation particle counter (CPC, model 3772,

157 TSI Inc., USA). The SMPS composed of a DMA (model 3081A, TSI Inc., USA) and a CPC (model 3775,

158 TSI Inc., USA) was employed to measure particles in a size range of 10-800 nm. The APS was used to

159 measured submicron particles ranging from 0.5 μm to 10 μm.

160 Size-resolved CCN activity was measured with a combination of a cloud condensation nuclei

161 counter (CCNc, model 200, DMT Inc., USA) and another SMPS. The CCNc-200 has two parallel cloud

162 columns, which can be used to measure the CCN concentration ($N_{CCN}$) simultaneously. The

163 supersaturation of each column was set to be 0.1%, 0.2% and, 0.4%, 0.7%, 0.9%, and 1.0%, respectively.

164 The dry particles were firstly neutralized by an X-ray neutralizer (model 3088, TSI Inc., USA) and were

165 then classified by a DMA (model 3081A, TSI Inc., USA). The monodisperse particles were split into

166 three streams: two to the CCNc for measurement of the $N_{CCN}$ (with a flow rate of 0.6 LPM) and one to

167 the CPC for measurement of total particle number concentration ($N_{CN}$, with a flow rate of 0.3 LPM).

168 Prior to the campaign, the SMPSs was calibrated with standard polystyrene latex spheres (PSL, with a

169 size of 20, 50, and 200 nm) and the CCNc-200 was calibrated with ammonium sulfate (($NH_4$)$_2$)$SO_4$

170 particles at the six SSs (0.1%, 0.2%, 0.4%, 0.7%, 0.9%, and 1.0%).

171 **2.2.2 Aerosol hygroscopicity measurement**

172 Hygroscopicity of atmospheric particle at various size ranges was measured by a hygroscopic

173 tandem differential mobility analyzer (HTDMA), consisted of two DMA (model 3081L, TSI Inc., USA),


a Nafion humidifier (model MD-700, Perma Pure Inc., USA), a heated tube and a condensation particle
counter (model 3788, TSI Inc., USA). The dry particles were firstly neutralizer by an X-ray neutralizer
(model 3088, TSI Inc., USA) and subsequently were classified by a DMA for six sizes in this study (30,
50, 80, 100, 150, and 200 nm). The selected particles at a specific diameter ($D_0$) were then introduced
into a humidifier under a fixed RH (90% in this study). Another DMA and a CPC were used to measure
size distribution of humified particles ($D_{wet}$).
**2.3 Methodology**
**2.3.1 Estimation of hygroscopicity based on the measurements**

The size-resolved activation ratio (AR) could be obtained from the measured $N_{CN}$ and $N_{CCN}$ by the

SMPS and CCNc-200 system and was inverted based on the method described by Moore et al. (2010).
The AR was then fitted with the sigmoidal function with respect to particle diameter $D_p$,

$$\frac{N_{CCN}}{N_{CN}} = \frac{B}{1+(\frac{D_p}{D_{50}})^C}$$     (1)

where $B$, $C$, and $D_{50}$ are fitting coefficients. The $D_{50}$ represents the critical diameter at which half of
the particles are activated at a specific SS.

The hygroscopic parameter κ can be obtained from the critical supersaturation (Sc) and the $D_{50}$

(Petters and Kreidenweis, 2007) by

$$\kappa = \frac{4A^3}{27D_{50}^3(\ln Sc)^2} \text{ , where } A = \frac{4\sigma_{s/a}M_w}{RT\rho_w}$$     (2)

where $\sigma_{s/a}$ is the surface tension of the solution/air interface and here it is temporarily assumed to be that
of pure water (0.0728 N m$^{-1}$ at 298.15 K), $M_w$ is the molecular weight of water (0.018 kg mol$^{-1}$), R is
the universal gas constant (8.31 J mol$^{-1}$ K$^{-1}$), T is the thermodynamic temperature in Kelvin (298.15 K),





and $\rho_w$ is the density of water (about 997.04 kg m⁻³ at 298.15 K).
The growth factor (GF) of selected particles can be calculated according to the following equation,
$$Gf = \frac{D_{wet}}{D_0}$$ (3)
In addition to the hygroscopic parameter calculated based on the SMPS and CCNc-200 system, the κ can
also be calculated from HTDMA measurement based on the growth factor,
$$\kappa = (Gf^3 - 1)\left[\frac{1}{RH}\exp\left(\frac{4\sigma_{s/a}M_w}{RT\rho_w D_0} - 1\right)\right]$$ (4)
Due to the effect of DMA diffusing transfer function, the TDMAfit algorithm (Stolzenburg and McMurry,
2008) was applied to narrow the uncertainty and fit the growth factor probability density function (GF-
PDF). Detailed data inversion process can be found elsewhere in Tan et al. (2013).

**2.3.2 Estimation of H₂SO₄ concentration and its contribution to particle growth**
The daytime gas phase H₂SO₄ concentration is estimated according to the proxy presented by Petäjä
et al. (2009),
$$[H_2SO_4] = \frac{k \cdot [SO_2] \cdot [OH]}{CS}$$ (5)
where $k$ is the reaction rate constant and is assumed to be 8.5 $\times$ 10⁻¹³ cm³ molecule⁻¹ s⁻¹ in this study
(Chen et al., 2014;Wang et al., 1988;Vignati et al., 2004), $[SO_2]$ is the concentration of SO₂ in molecules
cm⁻³, $[OH]$ is the concentration of OH radical in molecules cm⁻³, and the CS is the condensation sink in
s⁻¹ and it can be calculated from following equation,
$$CS = 2\pi D \sum_{Dp_i=Dp_{min}}^{+\infty} \beta_m N_i$$ (6)
where $D$ is the diffusion coefficient of the H₂SO₄ vapor (assumed to be 0.8× 10⁻⁵ m² s⁻¹ in this study),
$\beta_{m,i}$ is the transitional regime correction factor which can be calculated from the Knudsen number





(Fuchs and Sutugin, 1971), and $N_i$ represents the particle number concentration at $Dp_i$.

Framework for 0-D Atmospheric Modeling (F0AM) v3.1(Wolfe et al., 2016) is a zero-dimensional

atmospheric box model which was used to simulate the concentration of OH radical in the atmosphere.
The model was constrained with a set of online measured trace gases, VOCs, and meteorological data.
The employed chemical mechanism is Master Chemical Mechanism (MCM) v3.3.1. More detailed
description of model setup can be found in Wang et al. (2020).

The required vapor concentration of $H_2SO_4$ ($C_{v,GR=1\ nm\ h^{-1}}$) for a growth rate of 1 nm h$^{-1}$ in a

certain particle size range ($D_{p,initial}$ to $D_{p,final}$) can be calculated from the following equation,
$$C_{v,GR=1\ nm\ h^{-1}} = \frac{2\rho_v d_v}{\alpha_m m_v \Delta t} \cdot \sqrt{\frac{\pi m_v}{8kT}} \cdot \left[ \frac{2x_1+1}{x_1(x_1+1)} - \frac{2x_0+1}{x_0(x_0+1)} + 2ln\left(\frac{x_1(x_0+1)}{x_0(x_1+1)}\right) \right]  \qquad (8)$$
where $\rho_v$, $m_v$ and $D_v$ is the density, mass and diameter of $H_2SO_4$, which was assumed to be 1830 kg
m$^{-3}$, 98 amu, and 0.55 nm, respectively (Nieminen et al., 2010;Jiang et al., 2011), $\alpha_m$ is the mass
accommodation coefficient (assumed to be unity in this study), $x_1$ and $x_0$ are the ratios of $D_v$ to
$D_{p,final}$ (10 nm in this study) and $D_{p,initial}$ (3 nm in this study), $\Delta t$ (in s) is the time for particle growth
from $d_{p,initial}$ to $d_{p,final}$ ($\Delta t = \frac{d_{p,final}-d_{p,initial}}{GR}$) with a growth rate of 1 nm h$^{-1}$, and $k$ is the
Boltzmann constant (1.38×10$^{23}$ J K$^{-1}$).

Thus, the growth rate contributed from condensation of $H_2SO_4$ vapor can be obtained,

$$GR_{H_2SO_4} = \frac{[H_2SO_4]}{C_{v,GR=1\ nm\ h^{-1}}}  \qquad (9)$$
The average calculated $H_2SO_4$ concentration during particle growth can be calculated using Eq. (5). The
resultant $GR_{H_2SO_4}$ can be overestimated because the assumption of unity for $\alpha_m$ in Eq. (8) is not
necessary the case because not all $H_2SO_4$ molecules end up loss for their collisions with pre-existing
particles.



### 2.3.3 Estimation of growth rate (GR) and formation rate (J)

The observed particle growth rate (GR) is defined as the diameter change of nucleated particles
($dDp_{nuc}$) for a time period (dt),
$$GR = \frac{dDp_{nuc}}{dt} \tag{10}$$
Here log-normal distribution function method was adopted and the PNSD was fitted to obtain the
representative diameter for nucleated particles during NPF events (Kulmala et al., 2012),
$$\frac{dN}{dlogD_p} = \frac{N}{\sqrt{2\pi\sigma}} exp\left(-\frac{ln^2(\frac{Dp}{Dp_{gmd}})}{2\sigma}\right) \tag{11}$$
where $D_p$ is particle diameter, N is total particle number concentration, $Dp_{gmd}$ is geometric mean
particle diameter and it was also used as the representative particle size in Eq. (10). In this study, the
PNSD was found to have a significant mode in a size range of 3- 60 nm during NPF events and we hence
applied one log-normal mode fitting. At each time step, the PNSD was fitted using Eq. (11) and the
$Dp_{gmd}$ as a function of time, that is, the growth rate, was determined according to Eq. (10).
The formation rate ($J_k$) described the flux through a certain diameter (k) during NPF events and it
is calculated based on the formula given in Cai and Jiang (2017),
$$J_k = \frac{dN_{[Dp_k,Dp_u)}}{dt} + \sum_{Dp_g=Dp_k}^{Dp_{u-1}} \sum_{Dp_i=Dp_{min}}^{+\infty} \beta_{(i,g)} N_{[Dp_i,Dp_{i+1})} N_{[Dp_g,Dp_{g+1})} -$$
$$\frac{1}{2} \sum_{Dp_g=Dp_{min}}^{Dp_{u-1}} \sum_{Dp_i^3=max(Dp_{min}^3,Dp_k^3-Dp_{min}^3)}^{Dp_{i+1}^3+Dp_{g+1}^3 \leq Dp_u^3} \beta_{(i,g)} N_{[Dp_i,Dp_{i+1})} N_{[Dp_g,Dp_{g+1})} + n_u \cdot GR_u \tag{12}$$
where $N_{[Dp_k,Dp_u)}$ is particle number concentration in a size range from $Dp_k$ to $Dp_u$ (exclude particles
with diameter $Dp_u$), $Dp_k$ and $Dp_u$ are the lower and upper bound diameters (here 3 and 30 nm
respectively), $\beta_{(i,g)}$ is the coagulation coefficient for collisions between particles with diameter $Dp_i$
and particles with diameter $Dp_g$, $n_u$ is the particle distribution function at $Dp_u$ and $GR_u$ is the
growth rate calculated using Eq. (10) at $Dp_u$. Note that the calculation of formation rate using Ep. (12)





257 is based on two assumptions: (1) Dilution and other particles sources and losses except for coagulation

258 loss in the size range from $Dp_k$ to $Dp_u$ are negligible; (2) Net coagulation of particles is negligible.

259 **2.3.4 Measurement based NPF simulations**

260 For a regional NPF event, the evolution of particle size distribution is governed by the population

261 balance equations (Lehtinen et al., 2003; Kuang et al., 2012):

262 $$\frac{dN_{k^*}}{dt} = J_{k^*} - GR \cdot n_{k^*} - N_{k^*} \sum_{Dp_i=Dp_{min}}^{+\infty} \beta_{(k^*,i)} N_i \tag{13-1}$$

263 $$\frac{dN_k}{dt} = GR \cdot n_{k-1} - GR \cdot n_k + \frac{1}{2}\sum_{Dp_i=Dp_{min}}^{k-1} \beta_{(i,\varphi)} N_i N_\varphi - N_k \sum_{Dp_i=Dp_{min}}^{+\infty} \beta_{(k,i)} N_i \tag{13-2}$$

264 $$Dp_\varphi^3 = Dp_k^3 - Dp_i^3 \tag{13-3}$$

265 In the equations, class $k^*$ represents the smallest stable particle (here 3 nm particles), $J_{k^*}$ is the

266 formation rate calculated using Eq. (12). Class $k$ represents the particles with diameter $Dp_k$. The first,

267 second, and third terms on the right-hand side (RHS) of Eq. (13-1) represent the formation, condensation,

268 the coagulation sink terms, respectively. The first and second terms, the third, and fourth terms on the

269 RHS of Eq. (13-2) represent the condensation growth terms, a coagulation source (CoagSrc) term, and

270 the coagulation sink (CoagSnk) term, respectively.

271 For a specific NPF event, the evolution of PNSD with a size range of 3-1000 nm was simulated

272 based on Eq. (13) using Matlab (version 2016a, Mathworks, Inc.). In the simulation, the background

273 particle distribution was assumed to be the average PNSD before 6:00 LT, the growth rate and formation

274 rate were the measured values obtained from Eq. (10) and Eq. (13), respectively, and the time step was

275 set to be 10s. The simulation is based on following assumptions: (1) The dynamics of newly-formed

276 particles are driven by coagulation and condensation. The influences of transportation, primary emissions,

277 dilution, and particle evaporation are negligible. (2) The influence of coagulation on the preexisting



particles is negligible. (3) The particle growth rate for all particle sizes is assumed to be the same at a
time during NPF events.
**3      Results and discussion**
**3.1 New Particle Formation (NPF) events at the Heshan Site**

A total of 20 NPF events were observed during this seven-weeks long field campaign. Here we

selected a typical event (29[th] October, 2019) for further investigation. As shown in Fig. 1a, new particle
formation occurred at about 9:50 Local Time (LT) when a significant concentration of 3-10 nm particles
were observed. Subsequently, continuous and steady growth of the newly-formed particles was observed
until the particles grew to about 70-80 nm at about 20:00 LT. The blue dots in Fig.1a represent the $Dp_{gmd}$
of nucleated particles and the red line represents the linear fitting, leading to an estimated growth rate of
8.0 nm h$^{-1}$. Prior to the event (around 9:50), the total particle number concentration ($N_{CN}$) remained low
(a concentration slightly below 10000 cm$^{-3}$) and rapidly increased when NPF event occurred, and then
reached its peak (about 56000 cm$^{-3}$) at 11:15 LT and subsequently decreased to 20000 cm$^{-3}$ at about 15:00
LT, and remained at this concentration for the rest of the day. A steady north wind was observed before
18:00 LT and shifted to northwest afterwards (Fig. 1c). The shift of wind direction led to change of air
mass as seen from the PNSD, leading to a sudden increase of the $N_{CN}$ at 18:00 LT (Fig. 1a and b). The
CCN concentration ($N_{CCN}$) at 1.0% SS increased from 5000 cm$^{-3}$ at around 10:00 to 11000 cm$^{-3}$ at about
15:00 LT, when the nucleated particles grew to the CCN size (Fig. S1). The $D_{50}$ at 1.0% SS was apparently
the smallest critical diameters among all the SSs, the size that was easily reached during NPF and was
significantly affected by the newly-formed particles, we thus only discussed the variation of the $N_{CCN}$ at



1.0% SS in the following section. The sudden increase of $N_{CCN}$ at 18:00 LT could be attributed to change
of the air mass due to transportation, consistent with the changes of the PNSD, the $N_{CN}$, and wind
direction (Fig. 1a-c). The activation ratio (AR) was about 0.5 before dawn and dropped to about 0.2 just
prior to the event (Fig. 1b). This ratio continued to decrease to its trough at the time corresponding to the
maximum of $N_{CN}$ and then increased again to about 0.6 at 15:00 LT during particle growth, slightly higher
than the value before dawn. Clearly, NPF can not only add a large number of particles to the atmosphere
but also increase the $N_{CCN}$ and AR after particles are formed and grow. The wind speeds were about 3 m
$s^{-1}$ during initial formation and growth, and decreased to about 1.5 m $s^{-1}$ during most of the particle
growth periods.
Formation of gaseous $H_2SO_4$ was favored by intensive photochemistry. Significant j-values of $O(^1D)$
(in $s^{-1}$) were observed during the day (from about 7:00 to 17:00) with a maximum value of $2\times10^{-5}$ $s^{-1}$ at
noon and symmetrically distributed before and after noon. The average calculated concentration of
$H_2SO_4$ during particle formation (10:00-12:00 LT) was about $1.9\times10^7$ $cm^{-3}$, about an order higher than
that (about $7 - 12\times10^6$ $cm^{-3}$) in a mountain region in Germany (Wu et al., 2013a) and close to that (about
$2-5\times10^7$ $cm^{-3}$) in a rural region of Sichuan in China (Chen et al., 2014). Considering an uncertainty of
40% in estimation of $H_2SO_4$ concentration (Wu et al., 2013b), the GR contributed by condensation of
gaseous $H_2SO_4$ only was about 0.78-1.12 nm $h^{-1}$, or about 5.6% -20.0% of the total observed particle
growth rate in a size range of 3-10 nm. This implies that other compounds (e.g., organic vapors) than
$H_2SO_4$ play significant roles in the growth process of newly-formed particles which was widely reported
in literatures (Boy et al., 2005;Casquero-Vera et al., 2020;Paasonen et al., 2010).





**3.2 The impact of hygroscopicity and surfactants on $N_{CCN}$**
The ability that atmospheric particles can serve as CCN is determined by several factors including
sizes, chemical composition, surface tension, and water saturation ratio of the particles (Farmer et al.,
2015). The organic matter in particles can act as surfactants to lower the surface tension of the particles
and hence can increase the CCN activity (Ovadnevaite et al., 2017). Previous studies showed that the
presence of surfactants led to discrepancies of κ values between measurements using different techniques
under sub-saturation (HTDMA measurements) or supersaturation conditions (CCNc measurements) (Cai
et al., 2018; Wex et al., 2009; Rastak et al., 2017). Figure 2 compares the κ values measured from several
locations including Heshan (this study, rural), Panyu (urban PRD, Cai et al., 2018, 2019), and South
China Sea (Cai et al., 2020). The median κ values measured by HTDMA in this study ranged from 0.1
to 0.18 in a size range of 30-200 nm, similar to those of particles primarily composed of organics (Deng
et al., 2018;Liu et al., 2018;Pajunoja et al., 2015), implying that chemical composition of the measured
particles was dominated by organics. In particular, the κ values measured using HTDMA ($\kappa_{HTDMA}$) in this
study were significantly lower than those from other studies. The κ values in a range of 0.21-0.31 were
reported for urban PRD and suburban North China Plain, which were likely attributed to high fractions
of water-soluble organic matters and inorganic compounds from traffic and industry emissions. The κ
values measured using CCNc ($\kappa_{CCN}$) fall in a range from 0.19 to 0.46, much higher than those from
measurements using HTDMA in this study. The discrepancy of the κ $\kappa_{HTDMA}$ and $\kappa_{CCN}$ values suggests
that surfactant effects could play an important role in CCN activation under sub-saturation and
supersaturation environments. Previous studies have shown that the organics in particles could lower
surface tension by about 0.01-0.032 N m$^{-1}$ (Ovadnevaite et al., 2017; Liu et al., 2018; Engelhart et al.,





2008; Cai et al., 2018), leading to the decrease of the $D_{50}$ and higher $\kappa$ values.

A new surface tension ($\sigma_{s/a}^*$=0.060 N m$^{-1}$) was adopted to calculate the $\kappa_{CCN}$ using Eq. (2) based on

the measured critical diameter ($D_{50}$), which brought the $\kappa_{CCN}$ values at SS=1.0% and 0.9% within those
of $\kappa_{HTDMA}$, although the $\kappa_{CCN}$ values with this new $\sigma_{s/a}^*$ at other SSs were still higher, implying that the
surface tension is dependent on particle diameter. Surfactants can lower the $D_{50}$ of the particle which
then facilitates its activation as CCN. For particles with the same $\kappa$ value, the measured $D_{50}$ by fitting of
$N_{CCN}/N_{CN}$ using Eq. (1) was lower than the calculated value based on pure water surface tension using
Eq. (2) due to the surfactant effect. In order to estimate the impact of surfactant on particle activation,
the $D_{50}$ was recalculated using the surface tension of pure water (0.072 N m$^{-1}$) by Eq. (2) based on the $\kappa$
value from the CCN measurements with a surface tension correction (refer to $\kappa_{CCN}$ $\sigma_{s/a}^*$ and $\sigma_{s/a}^*$=0.060
N m$^{-1}$ in Fig. 2). We termed the above recalculated $D_{50}$ as the $D_{50}$ $\sigma_{s/a}$ to illustrate the surfactant effects
on the CCN activity during NPF events. Figure 3 shows the variation of the recalculated $D_{50}$ (here
$\sigma_{s/a}$=0.072 N m$^{-1}$) and the measured $D_{50}$ , along with the $Dp_{gmd}$ of the nucleated particles during the
NPF event. The measured $D_{50}$ was lower than the recalculated $D_{50}$ by about 10 nm. As a result, the
$Dp_{gmd}$ reached the measured $D_{50}$ at about 15:00 LT, about two hours earlier than it arrived at the
recalculated $D_{50}$, which indicates that the surfactant effects could lead to earlier activation of the newly-
formed particles as CCN. The earlier the $Dp_{gmd}$ reaches the critical diameter $D_{50}$, the higher the $N_{CCN}$
is because more particles can survive from being scavenged by preexisting particles. The difference of
PNSD at the time when the $Dp_{gmd}$ reached respectively the measured $D_{50}$ and the recalculated $D_{50}$ was
shown in Fig. 1S. The peak value of PNSD at 15:00 LT was about 20000 cm$^{-3}$ higher than the value at
17:15 LT. The $N_{CCN}$ also shows a difference (Fig. 4a).

We also investigate the effect of the surface tension on the $N_{CCN}$ at SS=1.0% by varying the value



of the surface tension. As we mentioned in the beginning of this section, a surface tension of 0.060 N
m$^{-1}$ ($\sigma_{s/a}^{*}$) was adopted when discussing the CCN activation at 1.0% SS and we assume that the
recalculated $D_{50}$ was based on this surface tension value. The average $D_{50}$ was the mean of the measured
and recalculated $D_{50}$. The $N_{CCN}$ is calculated by integrating particle concentrations above $D_{50}$ using the
following equation,
$$N_{CCN} = \int_{D_{50}}^{\infty} n_i dlogDp_i \qquad (14)$$
where $n_i$ is the particle distribution function at $Dp_i$. The $D_{50}$ can be the measured or recalculated one.
It was shown that the $N_{CCN}$ at SS=1.0% from integration of particles above the recalculated $D_{50}$ was
significantly lower than that above the measured $D_{50}$ after 12:00 LT (two hours after the occurrence of
the NPF event), with concentration differences of about 3000-4000 cm$^{-3}$ (Fig. 4a). The AR based on the
recalculated $D_{50}$ reached its minimal values between 10:00 and 12:00 LT, and then steadily increased
until 22:00 and subsequently decreased. The AR based on the measured $D_{50}$ reached its minimal during
the same period as the AR from the recalculated $D_{50}$; however, it then rapidly increased until 16:00 and
the continuing increase of the AR was much slower until 22:00, and also subsequently decrease for the
last hour of the measurement (Fig. 4b). This different trend was likely attributed to the continuing growth
of the nucleated particles to the CCN size prior to 16:00. Here, we define the deviation of $N_{CCN}$ based on
the recalculated $D_{50}$ from that based on the measured $D_{50}$ to evaluate the impacts of the surface tension
(primarily due to the surfactant effects) on the $N_{CCN}$,
$$\delta_{N_{CCN}} = \frac{N_{CCN,m} - N_{CCN,r}}{N_{CCN,m}} \qquad (15)$$
where the $N_{CCN,m}, N_{CCN,r}$ represent the $N_{CCN}$ based on the measured $D_{50}$ and the recalculated $D_{50}$ or
average $D_{50}$. The $\delta_{N_{CCN}}$ of the recalculated $D_{50}$ was about 0.1 prior to the NPF event, and reached a
peak value of 0.4 at 14:00 LT, and then decreased steadily to 0.1 at 22:00 and remained unchanged for


the last hour of the measurement (Fig. 4c). The results suggests that the decrease of the surface tension
due to the surfactant effects could lead to about 10% increase of the $N_{CCN}$ at 1.0% SS for non-event
period and about 40% increase during the NPF event (Fig. 4c). Apparently, the surfactants have more
significant effects on $N_{CCN}$ during the NPF event period than during non-event period, as the difference
between the $\delta_{N_{CCN}}$ based on the recalculated $D_{50}$ and the average $D_{50}$ was significant only during the
event period (12:00-18:00 LT).

The hygroscopicity of newly-formed particles can have profound impact on the $N_{CCN}$ during the

NPF event. During the campaign, the minimum particle size of CCN activity measurement was about
40-45 nm, thus the hygroscopicity of this size range was used to present the property of newly-formed
particles. In general, the hygroscopic parameter κ values for particles with a size range of 40-45 nm were
significantly higher during the early event period than during the non-event and other event periods,
corresponding to much higher hygroscopicity during the early event period than during the non-event
and other event periods (Fig. S2a). The calculated $H_2SO_4$ concentration peaked at about 10:00-11:00 and
subsequently decreased to a low level (about $0.5 \times 10^7$ $cm^{-3}$) until 16:00, implying that the increase of
hygroscopicity was related to the condensation of $H_2SO_4$ vapors. It should be pointed out that the high κ
values during 10:00~12:00 LT did not represent the hygroscopicity of newly-formed particles which were
primarily composed of particles much smaller than 30-40 nm. Those newly-formed particles grew to
about 40-50 nm at 14:00-16:00 (Fig.1a and Fig.3) and their κ values were obviously lower than the
average ones, implying that the organic vapors could play an important role during growth of newly-
formed particle as discussed in Section 3.1. The decrease of hygroscopicity due to condensation of
organic vapors can lead to an increase of about 3-4 nm for the $D_{50}$, much smaller than the increase of
about 10 nm induced by the surfactant effect which reduces the surface tension as discussed before. The



results indicate that the surfactant effect may play a more important role than hygroscopicity in the $N_{CCN}$
because the surfactant effect can largely decrease the $D_{50}$ during the NPF event when the number
concentration of particles is dominant by Aitken mode.
**3.3 The impact of the dynamic processes on $N_{CCN}$**
As discussed in section 2.3.4, the dynamical processes for newly-formed particles during nucleation
events are governed by the population balance equation (Eq. (13)). Here, we build a MATLAB program
to model the NPF event using Eq. 13, with input parameters including background particle distribution,
growth rate and formation rate. Notice that the simulation is based on the aforementioned three
assumptions. Figure 5 shows the measured and modeled PNSD, $N_{CN}$, and $N_{CCN}$ at 1.0% SS. To be
simplified, the background particle distribution was assumed to be the average particle distribution before
6:00 LT. The modeled PNSD and $N_{CN}$ agree very well with the measured ones, except the model fell to
reproduce the abrupt change of PNSD and $N_{CN}$ between 18:00 and 22:00. As discussed in section 3.1,
this discrepancy was attributed to the change of the air mass by wind direction which was not considered
in the model. However, there are considerable discrepancies between the modeled and the measured
$N_{CCN}$. The measured $N_{CCN}$ at 1.0% SS increased steadily after the occurrence of the NPF event (at around
9:00 LT) due to formation of high concentration particles at a size range of 10-60 nm until around 19:00
and subsequently the $N_{CCN}$ dropped for the rest of the day. The model $N_{CCN}$ started to increase at about
14:15 LT and reached its maximum level at about 17:00 LT. The model fell to reproduce the increase of
the measured $N_{CCN}$ before 16:00, although the reasons corresponding to the discrepancy are still unknown.
The modeled peak value of the $N_{CCN}$ at 1.0% SS was about 12000 $cm^{-3}$, which agreed very well with the
measured one (11000 $cm^{-3}$). Again, the model fell to reproduce the increase of $N_{CCN}$ due to the change of



the air mass between 18:00 and 22:00.

The effects of variation (halving or doubling) of the growth rate, formation rate, and the background

PNSD on the $N_{CN}$ and $N_{CCN}$ were investigated to test the sensitivity of those parameters. Figure 6 shows
the comparison of the measured $N_{CN}$ and $N_{CCN}$ and the modeled one based on the half or doubling of
each tested parameter, respectively. As can be seen from Fig. 6a, the modeled $N_{CN}$ values based on the
double GR, the double formation rate and the half background PNSD were higher than the corresponding
measured values, respectively, and vice versa. Doubling of the formation rate lead to formation of more
new particles and the half background PNSD corresponds to a low coagulation loss with pre-existing
particles, resulting in production of more new particles in the simulation. Doubling of the GR resulted in
a higher concentration of particles, probably due to the significant increases of the coagulation source
(Fig. S3b), while small decreases for both of the coagulation sink and growth term were found (Fig. S3a
and Fig. S3d). Since the newly-formed particles can grow to larger sizes under a higher GR, the PNSD
of new particles would be broader (Fig. S4), which provides a wider "region" for the coagulation sources.
Doubling of the FR (J) resulted in the highest modeled $N_{CN}$ (about 90000 cm$^{-3}$) among all simulated
cases; however, the modeled $N_{CCN}$ based on a double J was only the second highest value (about 15000
cm$^{-3}$). The highest modeled $N_{CCN}$ (about 25000 cm$^{-3}$) was found to double the GR and moreover it peaked
earlier at about 14:00 LT (two hours earlier than the other cases). Similarly, the highest modeled AR
(about 0.82) was from doubling the GR and an earlier peak time was also found (Fig. S5). The above
results can be attributed to the following two possible reasons: (1) Doubling of the GR made newly-
formed particles grow faster to the $D_{50}$ which facilitated the survival of more particles from coagulation
scavenging; (2) The $N_{CN}$ became higher by doubling the GR. If newly-formed particle grew slowly, for
example, the decrease of the GR to a half value would result in growth of most particles to diameters



below that of the $D_{50}$, leading to the smallest change of the $N_{CCN}$ compared to other cases (Fig. S5). The
pre-existing background particles can serve as the coagulation sinks for newly-formed particles and
hence can prevent them from growing to the CCN sizes. For example, under the double background
PNSD condition, the $N_{CN}$ reached its peak of about 38000 $cm^{-3}$ at about 11:00 and quickly dropped
afterward. The newly-formed particles contributed about 3000 $cm^{-3}$ to the $N_{CCN}$, or an AR of about 0.45
at about 17:30 LT, an insignificant change compared to the value for the non-event period, implying that
under a high background particle concentration, NPF events a minor contribution to the $N_{CCN}$. Doubling
or halving of the FR resulted respectively in contribution of about 11000 and 5000 $cm^{-3}$ to the $N_{CCN}$;
however, the magnitude of contribution from variation of the FR was relatively lower than that from the
GR and the background PNSD.

Figure 7 shows the comparison of the itemized absolute and fractional contribution of coagulation

sink, coagulation source, GR and J to the $N_{CCN}$ for the above several scenarios (model, double GR, half
or double J, and half or double PNSD). Here, the individual contribution was integrated from the
corresponding term in Eq. (13) for all particle sizes from the initial time of the NPF event to the time
when the $N_{CCN}$ reached the peak concentration. As clearly shown in Fig. 7, the coagulation source term
plays an more important role in the $N_{CCN}$ (with a fraction of about 13%) for the double GR case than any
other cases. As discussed above, doubling of FR (J) and halving of PNSD led to similar $N_{CCN}$ peak values
(about 15000 and 13500 $cm^{-3}$, respectively); however, the dynamics processes for the two scenarios were
significantly different. For the double J case, the formation term contributed about 240000 $cm^{-3}$ to the
$N_{CCN}$, much higher than the half PNSD case, and the CoagSnk and CoagSrc terms were much higher
(about -260000 and 50000 $cm^{-3}$, respectively) than any other cases due to formation of high concentration
of newly-formed particles. Moreover, under the double J scenario, the fraction of CoagSnk term was





higher, while the CoagSrc term was lower than the half PNSD case, indicating a more significant
coagulation scavenging with preexisting particles. As a result, the $N_{CN}$ quickly dropped from its peak
value to a concentration level similar to the half PNSD case within one hour (Fig. 6a). Based on the
above reasons, the contribution of the newly-formed particles to the $N_{CCN}$ was relatively smaller for the
double J case than the double GR or half PNSD cases, although its coagulation source term and J term
were the highest among all the cases.
**3.4 Modeling of the impact factors on the $N_{CCN}$ during NPF events**

Here we include two more NPF events to investigate the influence of several important impact

factors (growth rate, formation rate, and background particles) on the $N_{CCN}$, one from this campaign
(October 18[th], 2019), another from the field campaign in Panyu (December 12[th], 2014). Both campaigns
were conducted in the PRD region, details of the field campaign in Panyu can be found in Cai et al.
(2018). We applied the same model to simulate NPF as discussed in the previous section. Figure 8 shows
the measured (a), modeled (b) PNSD, along with the $N_{CN}$ (c). For a better comparison among all the
cases, all the modeled PNSDs were based on the measured formation rate ($J_{10}$) due to lack of
measurement data for particles below 10 nm in the Panyu campaign. The background particle
distributions were assumed to be the average values before 7:00 LT. In addition, since no measurement
data were available for the CCN activity at 1.0% SS during the Panyu campaign, the $N_{CCN}$ for this
campaign was calculated from the average CCN activation curve at 1.0% SS in the two Heshan events
and the PNSD of the Panyu event using following equation,
$N_{CCN} = \int_{Dp_i = Dp_{min}}^{\infty} AR_i n_i d log Dp_i$                                    (16)
where the $AR_i$ is the average activation ratio (in Heshan) at $Dp_i$ and the $n_i$ is the particle distribution



function (in Panyu) at $Dp_i$.

In general, the modeled PNSDs agreed well with the measured ones for the NPF events under

investigation (Fig. 8a-f). The $N_{CN}$ values were excellently predicted during the initial particle formation
period before the maximum values were reached (Fig. 8g-i). In particular, the $N_{CN}$ was well predicted for
the study case (the October 29 event) except for the period when the air mass changed as has been
discussed in the previous section. For the October 18 event, however, the model began to underpredict
the $N_{CN}$ shortly after the $N_{CN}$ reached the peak value, while for the Panyu event (the December 12 event),
a significant underestimate (about 4100 cm⁻³ lower than the measured $N_{CN}$) for the peak concentration
was made at about 12:00 pm, due probably to the presence of a significant amount of other bigger
background particles (100-200 nm) after 12:00 pm which was not able to be taken into account in the
model (Fig. 8c). As a result, the predicted $N_{CCN}$ value was substantially lower than the measured one for
the December 12 event (Fig. 9c). This also indicates that the $N_{CCN}$ was primarily contributed from the
background preexisting particles rather than newly-formed particles form the NPF event in the December
12 event case. The maximum modeled peak $N_{CCN}$ value (about 7000 cm⁻³) is significantly lower that of
the other two events (about 15000 and 12000 cm⁻³, respectively), which could be attributed to the lower
growth rates, formation rate, and the high CS value (Fig. S6 for $J_{10}$ and table S1 for GR and CS). We
further simulate the December 12 event to investigate the most important impact factor that influences
the $N_{CCN}$ using different characteristics from the two other NPF events, including the growth rate on the
October 18 event (high growth rate scenario), the formation rate on the October 29 event (high formation
rate scenario) and the background PNSD on the October 29 event (mainly distributed in Aitken mode).
The results show that all the new modeled $N_{CN}$ value were higher than the initial modeled $N_{CN}$ value.
The $N_{CN}$ was significantly increased and earlier peaked (with a peak value about 38000 cm⁻³) under the



high formation rate scenario, while the $N_{CCN}$ was mainly affected and also earlier peaked under the high
growth rate scenario. The peak value of $N_{CCN}$ increased from 6000 cm$^{-3}$ to 14000 cm$^{-3}$ and the peak time
varied from 20:00 LT to 16:00 LT. The $N_{CN}$ value increased under the new background scenario; however,
the $N_{CCN}$ barely changed, implying that larger size particles in the preexisting background play a more
important role in scavenging newly-formed particles. We hence conclude that the newly-formed particles
with a higher growth rate would grow faster to the CCN size by avoiding higher number concentration
losses in the atmosphere (Fig. S7a). Our results highlight the importance of particle growth rate in
modulating the $N_{CCN}$ during NPF events.
**4    Conclusions**

Field measurements were conducted at a rural site in the PRD region of China during October and

November 2019. The contribution of new particle formation (NPF) to the $N_{CCN}$ was investigated based
on three chosen NPF events including two (29$^{th}$ October and 18$^{th}$ November, 2019) from this field
campaign and one (12$^{th}$ December, 2014) from a previous campaign in Panyu. The effects of several
controlling factors on the contribution were explored to better understand the CCN activation process.
These factors include formation rate, growth rate, background particle distribution, hygroscopicity and
surface tension of the particles. Significant discrepancies were found for the $\kappa$ values between
measurements under supersaturation (using CCNc) and those under sub-saturation (using HTDMA), due
partly to the pure water assumption for the surface tension when calculating the $\kappa$ values based on the
CCNc measurements. Organics in the particles could act as surfactants to lower the surface tension which
facilitate CCN activation during NPF events. The results show that a surface tension value of about 0.060
N m$^{-1}$ instead of 0.073 N m$^{-1}$ (pure water assumption) could decrease the $D_{50}$ (SS=1.0%) for 10 nm


particles, bringing the agreement of the $\kappa$ values between CCNc and HTDMA measurements. The
surfactant effects caused by organics in the particles would increase the $N_{CCN}$ at SS=1.0% by about 20%
during non-event periods and by about 40% during NPF events. In addition, an earlier peak time was
also observed because much higher number concentrations of small particles (3-100 nm) during the event
would lead to smaller $D_{50}$.

The dynamic population balance equations were employed to qualitatively simulate NPF events

under different case scenarios (coagulation term, formation term and growth term). Sensitivity studies
were then performed to analyze the contribution of each aforementioned term to the $N_{CCN}$. The results
show that high formation rates, high growth rates, and low background particle concentrations lead to
high total and CCN concentrations, although different mechanisms were attributed to the high $N_{CN}$ and
$N_{CCN}$. High formation rates lead to high particle production in the atmosphere; likewise, high growth
rates produce a broad distribution of new particles and further increase the coagulation sources, while
low background concentrations result in low coagulation scavenging with preexisting particles. Among
these controlling factors, the growth rate was found to have the most profound impact on the $N_{CCN}$,
because a faster growth for newly-formed particles resulted in growing these particles to the CCN sizes
in a shorter time before they were scavenged by preexisting particles. The $N_{CCN}$ (SS=1.0%) measured
from the chosen event on 12[th] December, 2014 was significantly lower than that from two other chosen
events, initially attributed to the low growth rate, low formation rate, and low background particle
concentration. Sensitivity tests were then performed under different scenarios (the highest growth rate
form the event on 18[th] October, the highest formation rate and the lowest CS from the event on 29[th]
October, respectively) with change of only one factor for each simulation. The results show that the peak
value of the modeled $N_{CCN}$ increased from 6000 to 14000 cm$^{-3}$ with the new applied growth rate, leading





to a similar value to that from the event on 18th October, while the modeled $N_{CCN}$ values were barely
affected under the two other scenarios. These results highlight the importance of the growth rate in the
contribution of the controlling factors to the $N_{CCN}$. We concluded that surface tension and growth rate
played a major role in the contribution of NPF event to the $N_{CCN}$. More work on the other NPF cases is
obviously needed in order to better understand the contribution to the $N_{CCN}$ and its impact on climate.

*Data availability.* Data from the measurements are available upon request (Bin Yuan via
byuan@jnu.edu.cn).

*Supplement.* The supplement related to this article is available online at xxx.

*Author contributions.* **MC, MS** and **BY** designed the research. **MC, MS, BY**, **SH, YP, ZW, BL and QS**
performed the measurements. **MC, BY, JZ, HT, FL, SH, HX, LL, YP, ZW, BL and QS** analyzed the
data. **MC, BY** and **JZ** wrote the paper with contributions from all co-authors.

*Competing interests.* The authors declare that they have no conflict of interest.

*Acknowledgements.* This work was supported by the Key-Area Research and Development Program of
Guangdong Province (grant No. 2019B110206001), the National Key R&D Plan of China (grant No.
2019YFE0106300, 2018YFC0213904), the National Natural Science Foundation of China (grant No.
41877302, 91644225, 41775117), Guangdong Natural Science Funds for Distinguished Young Scholar
(grant No. 2018B030306037), Guangdong Innovative and Entrepreneurial Research Team Program



(grant No. 2016ZT06N263), and Guangdong Province Key Laboratory for Climate Change and Natural
Disaster Studies (Grant 2020B1212060025).

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



FIGURE CAPTIONS
Figure 1. The PNSD (a), $N_{CN}$, $N_{CCN}$ and AR (b), wind speed and wind direction (c), $j_{O(1D)}$, and
concentration of calculated $H_2SO_4$ (d) during the NPF event on 29th October, 2019. The blue dots in (a)
represents the geometric mean particle diameter ($Dp_{gmd}$) and the red line represents the linear fitting.
Figure 2. The median and interquartile κ obtained from HTDMA and CCN measurements during this
campaign, at the Panyu site (urban Guangzhou), and from South China Sea. The κ was pointed against
the corresponding median $D_{50}$ (CCN measurement) or selected diameter (HTDMA measurement). Dots
represent the median values and bars represent the interquartile ranges. The κ values in this measurement
were obtained from HTDMA measurement (in bule) and CCNc measurement (ss=0.1%, 0.2%, 0.4%,
0.7%, 0.9%, and 1.0% in red and yellow for different surface tensions). The yellow lines and dots
represent the κ values recalculated based on $\sigma^{s/a*}$. The κ values in the Panyu measurement were obtained
from HTDMA measurement (in purple) and CCNc measurement (ss=0.1%, 0.2%, 0.4%, and 0.7%, in
green). The κ values from the South China Sea were obtained from CCNc measurement (ss=0.18%,
0.34%, and 0.59%, in light blue). The κ values from the North China Plain were obtained from HTDMA
measurement.
Figure 3. The variation of $Dp_{gmd}$ (blue dots), measured $D_{50}$ (yellow dots) and recalculated $D_{50}$ (red dots)
based on pure water surface tension.
Figure 4. The variation of $N_{CCN}$ (a), activation ratio(b), and $\delta_{CCN}$ (c) based on the measured $D_{50}$, the
recalculated $D_{50}$, and the average $D_{50}$. The red line represents the measured values. The yellow line
represents the values calculated based on the surface tension of pure water (0.072 N m⁻¹). The purple line
represents the values calculated from the average $D_{50}$. The green region represents the interquartile values
calculated from the interquartile $D_{50}$.



Figure. 5 The measured and model PNSD (a and b), $N_{CN}$ (c) and $N_{CCN}$ (c). The blue lines in (c) represent
the measured values and the red lines represent the model values.
Figure 6. The variation of measured and model $N_{CN}$ (a) and $N_{CCN}$ (b) at 1.0% SS. The simulations was
based on standard characteristic (red solid line), halving of GR, formation rate and background particle
distribution (orange, purple and green solid line, respectively) and doubling of GR, formation rate and
background particle distribution (orange, purple and green dash line, respectively).
Figure 7. The number contribution (a) and its fraction (b) of CoagSnk term, CoagSrc term, GR term, and
formation (J) term to the $N_{CCN}$ when it reached its peak value based under different case scenarios.
Figure 8. The measured PNSD (a, b, and c), model PNSD (d, e, and f), measured $N_{CN}$ and model $N_{CN}$ (g,
h, and i) during different NPF events. Solid and dash lines represent the measured and model $N_{CN}$,
respectively.
Figure 9. The measured and model $N_{CCN}$ (SS=1.0%) during different NPF events.
Figure 10. The measured and model $N_{CN}$ (a) and $N_{CCN}$ (b) on the Panyu NPF event. The bule line
represents the measured value. The red, yellow, purple and green lines represent the simulated $N_{CCN}$
based on standard input, growth rate of the NPF event on October 18[th], formation rate of the NPF event
on October 29[th], and new background particle distribution of the NPF event on October 29[th], respectively.






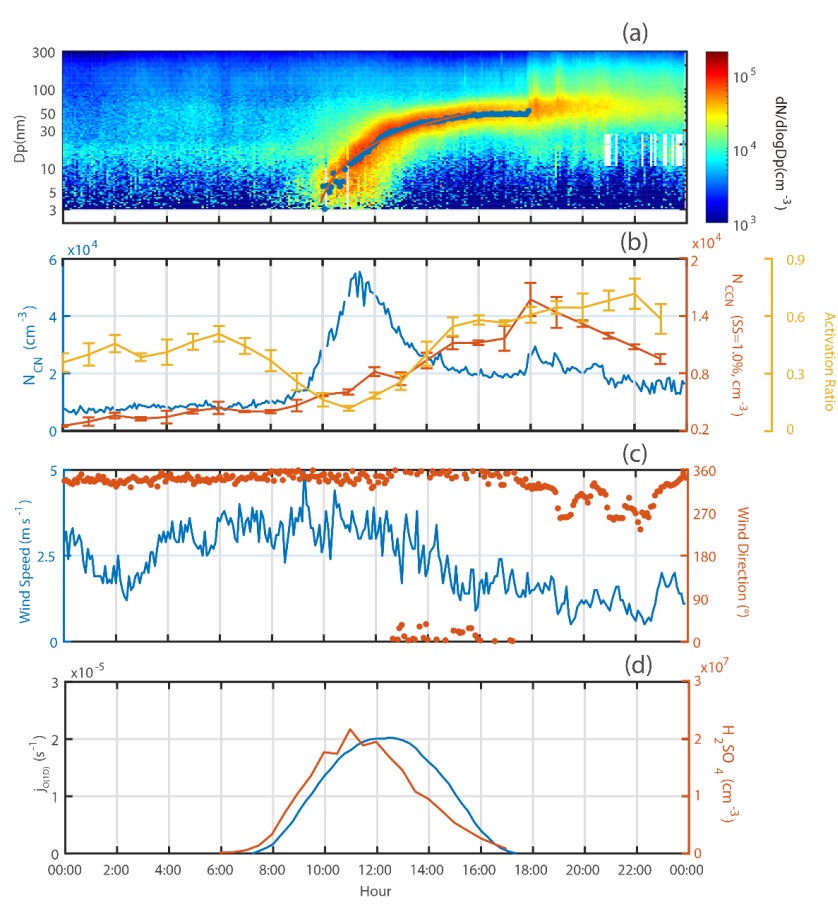


Fig. 1.





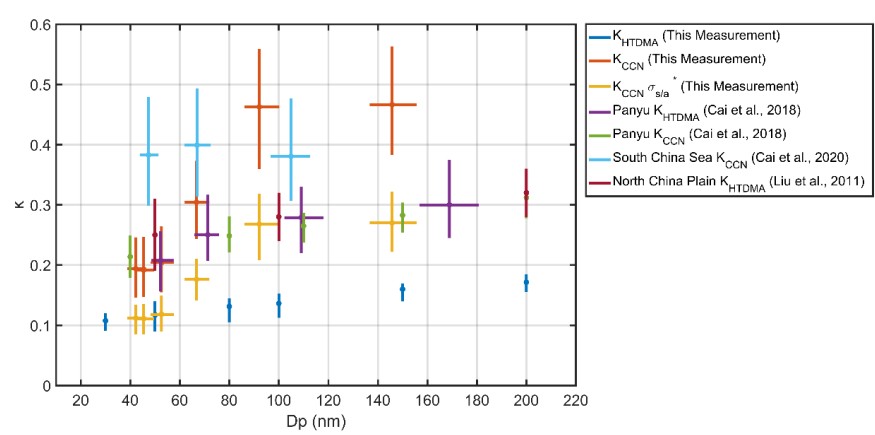



Fig. 2.





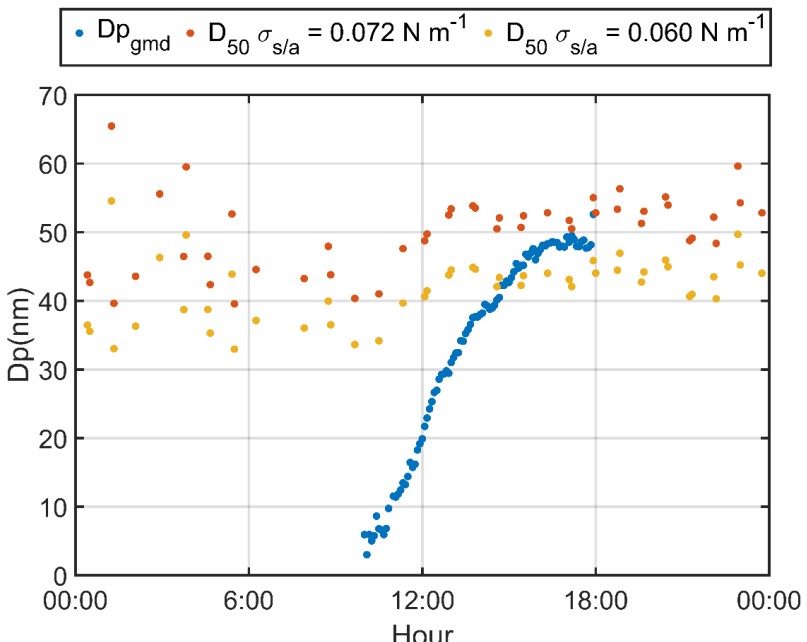



Fig. 3.





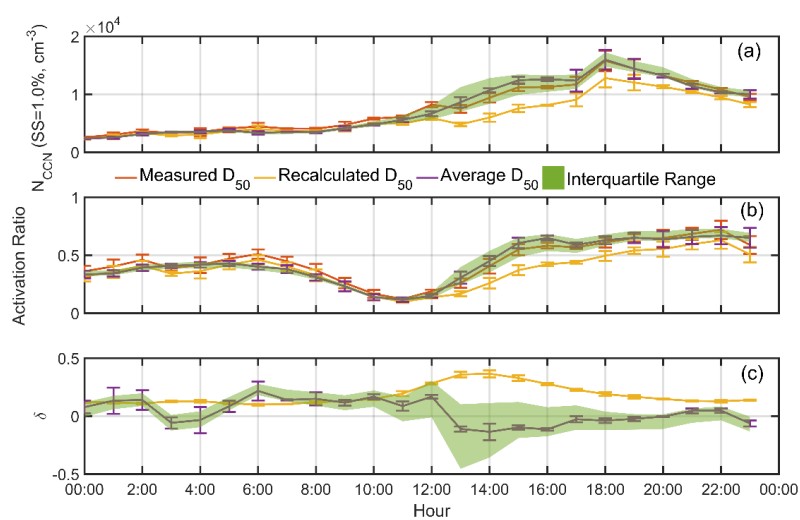



Fig. 4.





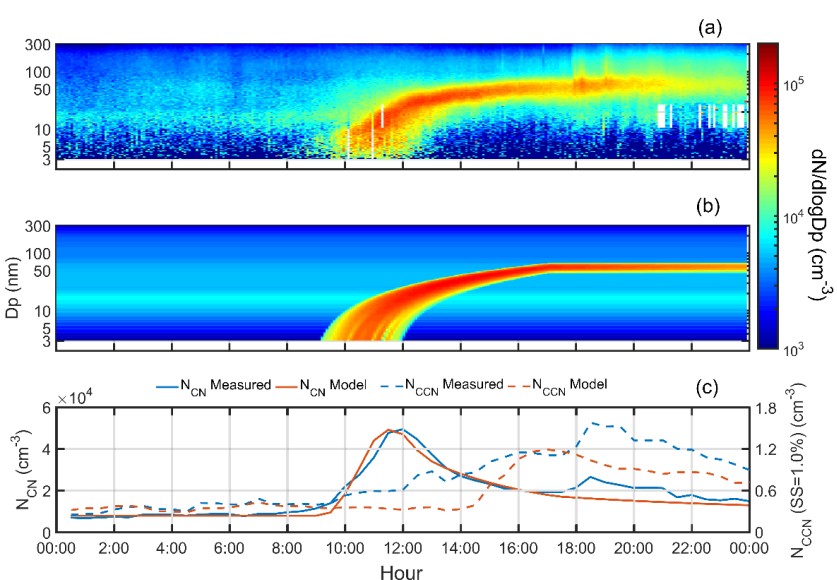



Fig. 5.





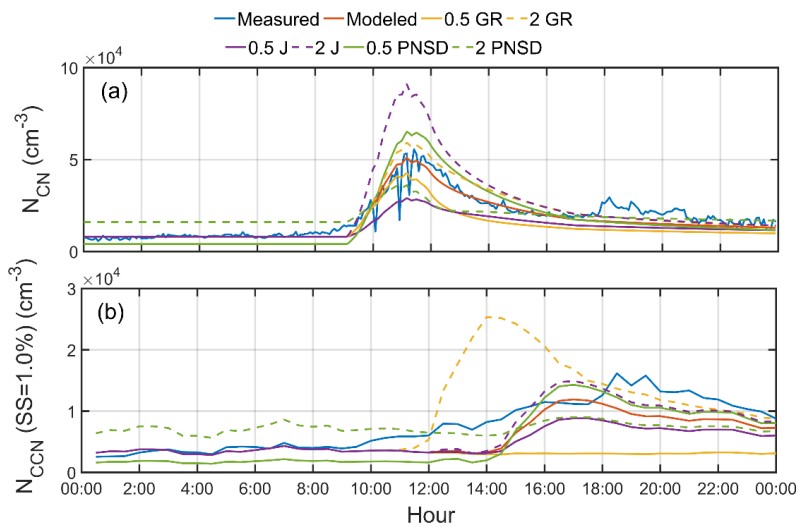



Fig. 6.





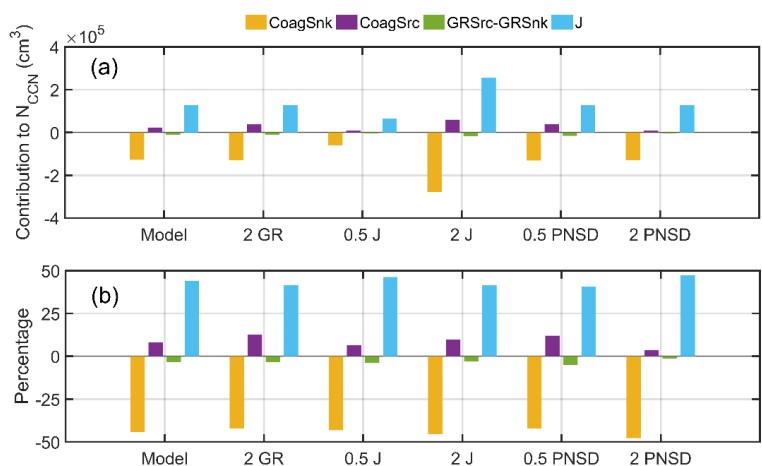



Fig. 7.





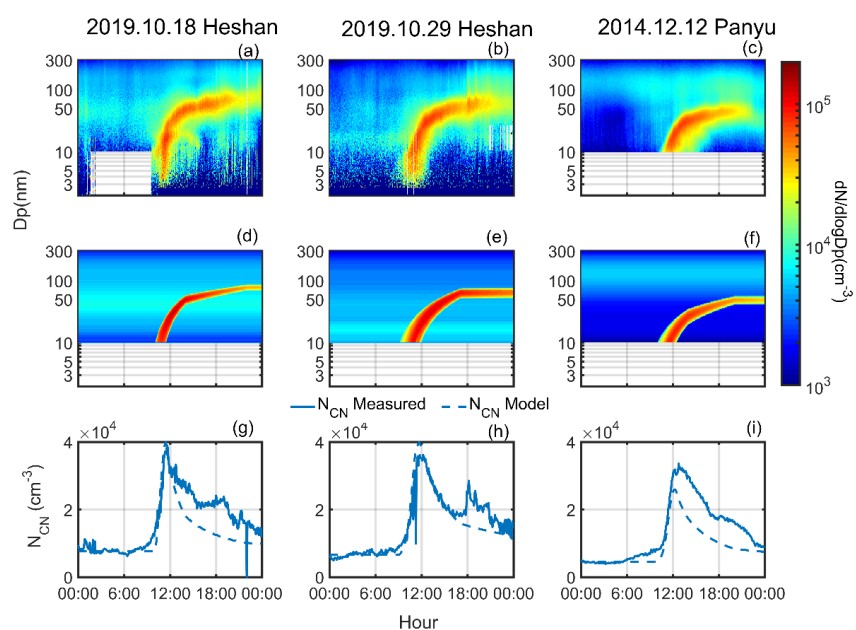



Fig. 8.





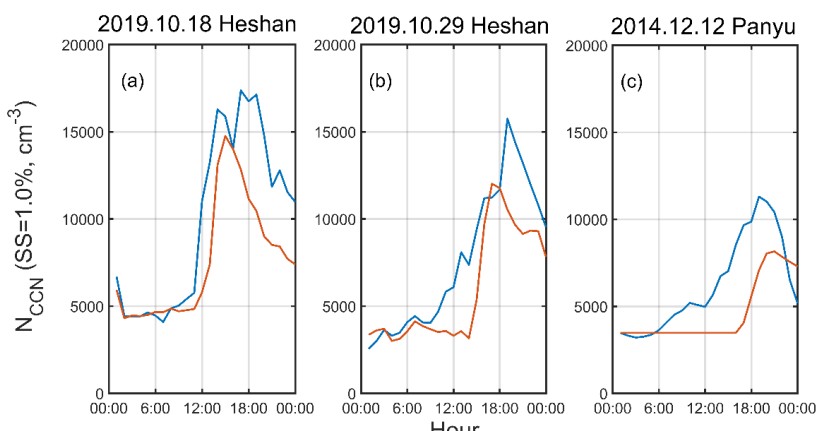



Fig. 9.



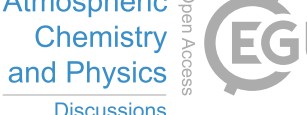

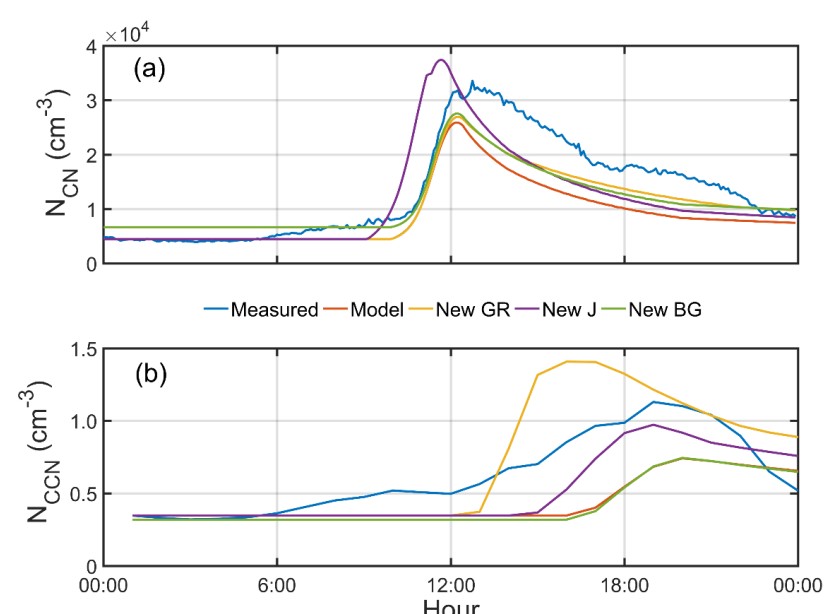



Fig. 10.
