# Peer review of "The important roles of surface tension and growth rate"

_Atmospheric Chemistry and Physics, 2020_

## Referee Comment (RC1) · Anonymous Referee #2 · 6 Jan 2021

This paper examines the effects of hygroscopicity, surface tension and aerosol processes on the NPF contribution to Nccn, based on field observations and modeling of three NPF events at a rural site in southern China. The study results and implications are of interest to ACP readers. The adopted experimental setup and methodology are well-established, comprehensive, and thus reasonable. The manuscript is generally well-written and organized, though some data presentation, interpretation and discussion can be improved.

[Figure]

My major comments, followed by minor comments are given below:

1. Because there are only three NPF events, quite thoroughly, analyzed in this study, it is crucial that the authors should somewhat discuss the representativeness of those three NPF events. The dominant mechanisms driving NPF vary with time and location.
2. Although the hygroscopicity and the estimated surface tension were derived under different water saturation ratio (undersaturated vs. supersaturated), two are interlinked with each other. The discussion in Section 3.2 seems to treat the two as unrelated factors. Also, e.g., in the abstract, the authors suggested the surface tension is more important than hygroscopicity (line 37). It is recommended that the authors elaborate/clarify on the rationale of discussion based on adjusting only the surface tension in kccn (but not kHTDMA?), or kHTDMA is the "reference" hygroscopic parameter, and the potential relationships between the two. The discussion and statements should be rephrased to accurately describe the observed cause and effect in a relationship.
3. In Section 3.2, lines 389-397, the use of the term "newly-formed" particles should be more specific and consistent, whether it refers to 40-45 nm particles, or « 30-40 nm particles. It is unclear that the k values discussed therein are kccn or kHTDMA. The "gradual" drop of sulfuric acid (SA) does not necessarily imply it is responsible for the increase of k values because SA condensation is considerably more favorable with larger pre-existing particles, and/or the consideration of oxidant availability. 4. Section 3.3 seems to be an add-on modeling analysis of the NPF events not strongly or quantitatively linked to the hygroscopicity and surface tension. The derived conclusions about formation/growth rates and coagulation loss are not new, but expected. This analysis is then extended to Section 3.4 where the three NPF events from two locations are compared. My concerns are (1) the modeled-Nccn deviate notably from measured-Nccn (Figs 5, 6 and 9), and (2) how the findings herein are related to the other subjects of interest regarding the hygroscopicity and surface tension. Please clarify. 5. With respect to surface tension, the authors are encouraged to review/include recent studies on the impact of morphology of organic/inorganic mixture on surface tension. As such, the discussion would be more in-depth and balanced.

[Figure]

Minor comments: 1. A schematic diagram of the experimental setup is recommended. 2. The lowest measurable particle diameter in this study is 1 nm. Is there any reason not to use this for the estimation of formation and growth rates, instead of 3 nm (lines 227, 253, 265)? 3. Line 415 and other instances, the "fail" is misspelled as "fell."
* * *

---

## Referee Comment (RC2) · Anonymous Referee #1 · 22 Jan 2021

Cai et al. present measurements of how new particle formation events and hygroscopicity impact cloud condensation nuclei concentrations. These observations were done in Guangdong Province at a rural site. The authors also modeled particle formation to see how it would impact CCN concentrations. The manuscript was at times difficult to understand due to frequent typos and unstructured paragraphs. However, the science done by the authors is sound as it has been implemented by numerous groups before. Also, their conclusions are logical. This paper fits ACP and should be published once

the authors address the below comments.

Major comments:

The authors used a DEG SMPS to measure the particle size number distribution but instead used formation rate at 3 nm and growth rate from 3-60 nm. Is there a reason the <3 nm bins were not used? I would think the growth rate from 1-3 nm would play an even larger role in their particle number concentration modeling done in section 3.3 Can the authors comment on this and maybe look into their 1-3 nm data to see how that would impact their data analysis?

Page 10, line 207: The authors did not directly measure sulfuric acid concentration but instead used a proxy based on SO2. How accurate is the proxy for the region they are measuring in? Every 1-2 years, a new proxy sulfuric acid paper is published from a new measurement location. Each of these proxy equations is different, with different parameters and different coefficients. See (Mikkonen et al., 2011). Why did the authors choose this proxy? Could they provide some gauge of uncertainty as a function of time? The authors say 40% (line 313) but how was this estimated and how does it vary with CS and OH concentrations? This would be especially useful as the authors compare trends of when sulfuric acid concentrations peak at specific times of the day compared to growth rate and when hygroscopicity increased (page 19 line 397).

Generally, the terminology used is confusing. Particle formation rate is used to described nucleation and of CCN (page 19 line 389). The convention is newly formed particles are small (<10 nm). It would be helpful if the authors could find a better phrase to call 30-40 nm particles. Also, measured, averaged, recalculated D50 are used. However, in the text, they often just say D50. Which D50 is it? Example (though not the only one) is page 19 line 403, page 20 line 406, etc. Maybe add a subscript to the D50 to make it clearer? Page 21 line 436: How does doubling the GR result in increasing particle concentration because of higher coagulation source? More coagulation would mean lower particle concentrations?

Page 21 paragraphs starting with line 427: This section is very repetitive and difficult to understand. What do the authors want us to take away from this section? Is there a more efficient way of communicating this information then just listing off every possible configuration of the model? The findings in this section are not new science so sticking to highlights of why this section is needed to convey the purpose of the paper would be helpful.

Minor comments: Line 27: environments and depend

Line 29: investigates

Line 38: than assuming pure water

Line 43 fact

Line 51: controlling factor is a weird phrase. What do you mean by controlling?

Line 56-60 these two sentences are wordy and difficult to understand

Line 61: controlling factors, again what do you mean by this?

Line 63: also marine?

Line 75: matter

Line 76: and more easily

Line 77: matter

Line 80: awkward statement of constraining an accurate quantification of the aerosol. . .

Line 82: NPF event is

Line 103: growth rates

Line 124: have reported

Line 135: that manipulate is awkward phrase.

Line 234: necessarily the case, also this entire sentence should be rewritten to be more clear

Line 265: represents the smallest detectable particle size. The smallest stable size is likely much smaller than 3 nm.

Line 357: D50 is shown

Line 359: shows a difference of what?

Page 20: all fell should be failed

Line 429: measured NCN and NCCN and the modeled one is awkward phrasing

Line 451: what is double background PNSD condition?

Line 454: NPF events a minor contribution, awkward phrasing

Line 547: profound impact, awkward phrasing

Line 758: linear fit

Line 772: space after activation ratio

Figure 2: is there a way to not use shades of the same color on this graph? The blues cannot be easily distinguished. Maybe adding symbols would help?

Figure 6: maybe helpful to write in the legend 0.5xGR and 2xGR, etc. to make it clearer.

Table S1: background particle distributions

Figure S2: Is average value during the campaign mean it was averaged over that time of day interval over the whole campaign?

Figure S7: What is it meant by "new" GR, formation rate, and background PSND?

Reference cited in this review Mikkonen, S., Romakkaniemi, S., Smith, J. N., Korhonen, H., Petäjä, T., Plass-Duelmer, C., Boy, M., McMurry, P. H., Lehtinen, K. E. J., Joutsensaari, J., Hamed, A., Mauldin III, R. L., Birmili, W., Spindler, G., Arnold, F., Kulmala, M. and Laaksonen, A.: A statistical proxy for sulphuric acid concentration, Atmospheric Chem. Phys., 11(21), 11319–11334, https://doi.org/10.5194/acp-11-11319-2011, 2011.

---

## Author Comment (AC1) · 12 Mar 2021

We would like to appreciate the reviewer for providing valuable comments on our manuscript, and we have carefully addressed these comments point-by-point as follows. Please find the response (in red) to each comment below.

**Referee comments:**

This paper examines the effects of hygroscopicity, surface tension and aerosol processes on the NPF contribution to Nccn, based on field observations and modeling of three NPF events at a rural site in southern China. The study results and implications are of interest to ACP readers. The adopted experimental setup and methodology are well-established, comprehensive, and thus reasonable. The manuscript is generally well-written and organized, though some data presentation, interpretation and discussion can be improved.

**Major comments:**

1. Because there are only three NPF events, quite thoroughly, analyzed in this study, it is crucial that the authors should somewhat discuss the representativeness of those three NPF events. The dominant mechanisms driving NPF vary with time and location.

Reply: We appreciate the reviewer for this valuable suggestion. We totally agree with the reviewer that the dominant mechanisms driving NPF may vary temporally and spatially. We add a discussion on the representativeness of the three NPF events in lines 566-579 in section 3.4 as follows,

"It should be noted that the three NPF events discussed in this study were generally "Class I" regional NPF events, for which the growth rate and formation rate could be obtained with high confidence (Dal Maso et al., 2005). Other types (i.e., Class II proposed by Dal Maso et al. (2005)) were not considered since their growth rates and formation rates are extremely difficult to be determined, leading to high uncertainties in model simulation of these events. In addition, we did not include the "transport" type of NPF events, for which new particles were formed somewhere else and then transported to the measurement site, because the model ignores the impact of transport. Some events belonging to "Class II" type and "transport" type were observed during the campaign (Fig. S10). For the "Class II" type (Fig. S9 a), the number concentration and diameter of the nucleation and Aitken mode particles vary significantly. For the "transport" type (Fig. S10 b), the concentration of 3-30 nm particles at 10:00-12:00 LT was much lower than that of 30-70 nm at 12:00-20:00 LT, indicating the impact of transport. Investigation on the contribution of other NPF types to the $N_{CCN}$ is needed in future studies. Moreover, this study only analyzed three NPF events as representatives of Class I type in the PRD region, and more field campaigns in other regions and seasons are also needed to identify the major impact factor.
"

[Figure]

Figure S10. The "Class II" type NPF event (a) and "Transport" type NPF event (b) observed on 9[th] September and 14[th] November during the Heshan Campaign, respectively.

2. Although the hygroscopicity and the estimated surface tension were derived under different water saturation ratio (undersaturated vs. supersaturated), two are interlinked with each other. The discussion in Section 3.2 seems to treat the two as unrelated factors. Also, e.g., in the abstract, the authors suggested the surface tension is more important than hygroscopicity (line 37). It is recommended that the authors elaborate/clarify on the rationale of discussion based on adjusting only the surface tension in kccn (but not kHTDMA?), or kHTDMA is the "reference" hygroscopic parameter, and the potential relationships between the two. The discussion and statements should be rephrased to accurately describe the observed cause and effect in a relationship.

Reply: We agreed that the impact of surface tension on the hygroscopicity growth under subsaturated should also be considered. We have recalculated the $\kappa_{CCN}$ and $\kappa_{HTDMA}$ by adjusting the surface tension and found that the $\kappa_{HTDMA}$ was slightly changed with the change of surface tension (Fig. S3). The $\kappa_{HTDMA}$ $\sigma_{s/a}$* was not changed considerably with $\sigma_{s/a}$*=0.060 N m$^{-1}$ and thus this value ($\sigma_{s/a}$*) was still adopted in the following discussion.

[Figure]

Figure S3. The median and interquartile $\kappa_{HTDMA}$ and $\kappa_{CCN}$. The red and yellow line represent the κ value calculated based on $\sigma_{s/a}*$ (0.060 N m$^{-1}$)

We have modified the discussion in section 3.2 in lines 334-338,
"Previous studies showed that surfactants could modify the ability of water uptake, leading to discrepancies of κ values between measurements using techniques under different water saturation conditions, e.g., sub-saturation (HTDMA measurements) or supersaturation (CCNc measurements) (Cai et al., 2018; Wex et al., 2009; Rastak et al., 2017; Ruehl and Wilson, 2014)."
and lines 363-370:
"A surface tension value ($\sigma_{s/a}^*$=0.060 N m$^{-1}$) was adopted to calculate both the $\kappa_{CCN}$ (denoted as $\kappa_{CCN}$ $\sigma_{s/a}^*$) and $\kappa_{HTDMA}$ ($\kappa_{HTDMA}$ $\sigma_{s/a}^*$) using Eq. (2) and Eq. (4), respectively. No significant changes of κ values (i.e., from 0.11 to 0.10 for 30 nm particles) were found from TDMA measurements (Fig. S3), while the κ values from CCNc measurements using this surface tension value ($\sigma_{s/a}^*$) were still lower than those using pure water assumption and the differences became larger with increasing particle sizes, implying that the surface tension is dependent on particle diameter. It also implies that the κ value was more susceptible to surfactants under supersaturation condition, which can lower the $D_{50}$ of the particle for facilitating CCN activation."

3. In Section 3.2, lines 389-397, the use of the term "newly-formed" particles should be more specific and consistent, whether it refers to 40-45 nm particles, or « 30-40 nm particles. It is unclear that the k values discussed therein are kccn or kHTDMA. The "gradual" drop of sulfuric acid (SA) does not necessarily imply it is responsible for the increase of k values because SA condensation is considerably more favorable with larger pre-existing particles, and/or the consideration of oxidant

availability.

Reply:

We thank the reviewer for the suggestion.

(1) The size range of "newly-formed" particles is difficult to define, owing to the continuous growth processes during the event. In order to avoid confusion, these particles were referred to as "newly grown particles", since they grew from newly-formed particles. We have modified corresponding sentences in lines 411-414:

"The hygroscopicity of newly-grown particles can have significant impact on the $N_{CCN}$ during the NPF event. During the campaign, the minimum particle size of CCN activity measurement was about 40-45 nm (at 1.0% SS), thus the hygroscopicity of this size range was used to present the property of newly-grown particles, when they grow up to this size range.", and line 418-423, "It should be pointed out that the high κ values during 10:00~12:00 LT did not represent the hygroscopicity of newly-grown particles which were primarily composed of particles much smaller than 30-40 nm. Those new particles grew to about 40-50 nm at 14:00-16:00 (Fig. 1a and Fig. 3) and their κ values were obviously lower than the average ones, implying that the organic vapors could play an important role during growth of new particle as discussed in Section 3.1."

and lines 430-431, "As discussed in section 2.3.4, the dynamical processes for new particles during nucleation events are governed by the population balance equation (Eq. (13))."

(2) The κ discussed in this section was only limited to the $\kappa_{CCN}$ measured at 1.0% SS because the time resolution of HTDMA measurement was low (about 4 hours). We agreed that condensation of gaseous $H_2SO_4$ might not be responsible for the increase of κ values, and other organics vapors (e.g., amines) could be a possible reason for the increasing hygroscopicity. Based on the above reasons, we deleted the sentences in line 395-397 "The calculated $H_2SO_4$ concentration peaked at about 10:00-11:00 and subsequently decreased to a low level (about $0.5 \times 10^7$ cm$^{-3}$) until 16:00, implying that the increase of hygroscopicity was related to the condensation of $H_2SO_4$ vapors.". We have also modified the discussion in lines 414-418:

"In general, the $\kappa_{CCN}$ values for 40-45 nm particles were significantly higher (corresponding to much higher hygroscopicity) during early event period than during non-event and other event periods (Fig. S4a). Hence, we adopted a minimum size range of 40-45 nm particles for CCN activity measurements (at about 1.0% SS) to represent typically growth of newly-formed particles to this size range during the campaign."

[Figure]

Figure S4. The diurnal variation of $\kappa$ (a) and $D_{50}$ (b) measured at 1.0% SS. The blue color represents the average value during the campaign. The red color represents the value during the NPF events.

4. Section 3.3 seems to be an add-on modeling analysis of the NPF events not strongly or quantitatively linked to the hygroscopicity and surface tension. The derived conclusions about formation/growth rates and coagulation loss are not new, but expected. This analysis is then extended to Section 3.4 where the three NPF events from two locations are compared. My concerns are (1) the modeled-Nccn deviate notably from measured-Nccn (Figs 5, 6 and 9), and (2) how the findings herein are related to the other subjects of interest regarding the hygroscopicity and surface tension. Please clarify.

Reply:

(1) There are two possible reasons for the deviation between the modeled $N_{CCN}$ and measured $N_{CCN}$. Firstly, our model does not consider transport and local primary emissions which may partly contribute to the deviation. For example, a significant mode peaking at about 100 nm was observed for the Panyu event, suggesting impact of air mass transport or local emissions. Secondly, we assume constant background particle distribution during NPF events, while actual background PNSD varies substantially from one event to another for the three chosen NPF events. Noticeably, significant variation of Aitken mode was observed for the two Heshan events, leading to failure of reproducing the concentration trend of 10-60 nm particle at the early event stage. To clarify, we have modified the discussion in lines 535-543,

"For the October 18 event, however, the model underpredicted the $N_{CN}$ shortly after it reached the peak value which can be attributed to significant variation of Aitken mode during the event. For example, the model failed to reproduce concentration trend of 10-60 nm particle at the early event

stage (Fig. 10a-b). For the December 12 event, the model underpredicted a significantly lower peak concentration (about 4100 cm$^{-3}$ lower) at about 12:00 pm than the measured one, due probably to presence of a significant amount of larger background particles (100-200 nm) which were not taken into account in the model (Fig. 9c and Fig. 10c). As a result, the N$_{CCN}$ was underpredicted in two Heshan events (fig. 10a- b), owing to the fluctuation of background particle distribution and unexplained increase of concentration of particles at a size range of 10-60 nm at the beginning of event."

(1) In section 3.3, we mainly discussed the relationship between the dynamic processes and the N$_{CCN}$. We found that both the NPF characteristics and the properties of newly-formed particles could influence the N$_{CCN}$. We added several sentences to discuss the impact factors on the N$_{CCN}$ in lines 501-514,

"To compare different impacts of the characteristics and properties of newly-formed particles, the N$_{CCN}$ was simulated through varying parameters of different characteristics (case 1, 4 and 7) and properties (case 2, 3, 5, 6, 8 and 9). The input parameters for different cases are shown in Table S1. For case 2, 3, 5, 6, 8 and 9 scenarios, the surface tension or hygroscopicity was adjusted to match similar N$_{CCN}$ values based on different NPF characteristics (case 1, 4 and 7, respectively). The results show that doubling GR produces the most significant impact on the N$_{CCN}$, and the surface tension ($\kappa$ value) was adjusted to 0.030 N m$^{-1}$(1.2) to have the same impact (Fig. 8a). Obviously, a $\kappa$ value of 1.2 for hygroscopicity is much higher than that of many inorganics, e.g., H$_2$SO$_4$ ($\kappa$=0.90, Topping et al., 2005) and NH$_4$NO$_3$ (0.58, Topping et al., 2005). Meanwhile, the surface tension was lower than the values (0.049-0.060) reported previously (Ovadnevaite et al., 2017; Engelhart et al., 2008; Cai et al., 2018). However, doubling GR value (16.0 nm h$^{-1}$) was reasonable and consistent with previous studies (Mönkkönen et al., 2005; Foucart et al., 2018; O'Dowd et al., 1999), suggesting significant contribution of GR to the growth. For doubling formation rate and halving PNSD, the modified surface tension and $\kappa$ values were minor (Fig. 8b and c)."

Table S1. The input parameters for Case 1-9.

$2\times$ and $0.5\times$ represent doubling and halving the parameters, respectively.

|        | GR          | J          | PNSD        | $\sigma_{s/a}$ (N m$^{-1}$) | $\kappa$   |
|--------|-------------|------------|-------------|-----------------------------|------------|
| Case 1 | $2\times$   | $1\times$  | $1\times$   | 0.0728                      | Measured   |
| Case 2 | $1\times$   | $1\times$  | $1\times$   | 0.030                       | Measured   |
| Case 3 | $1\times$   | $1\times$  | $1\times$   | 0.0728                      | 1.2        |
| Case 4 | $1\times$   | $2\times$  | $1\times$   | 0.0728                      | Measured   |
| Case 5 | $1\times$   | $1\times$  | $1\times$   | 0.065                       | Measured   |
| Case 6 | $1\times$   | $1\times$  | $1\times$   | 0.0728                      | 0.15       |
| Case 7 | $1\times$   | $1\times$  | $0.5\times$ | 0.0728                      | Measured   |
| Case 8 | $1\times$   | $1\times$  | $1\times$   | 0.067                       | Measured   |
| Case 9 | $1\times$   | $1\times$  | $1\times$   | 0.0728                      | 0.13       |

[Figure]

Figure. 8 The model $N_{CCN}$ based on different characteristics (doubling growth rate and formation rate, and halving background particle distribution) and particle properties. Different colors and markers represent case 1-9, respectively.
"

5.  With respect to surface tension, the authors are encouraged to review/include recent studies on the impact of morphology of organic/inorganic mixture on surface tension. As such, the discussion would be more in-depth and balanced.

Reply: We thank the reviewer for valuable suggestions. We have added discussion (also the references) on some recent studies to show the impact of liquid-liquid phase separation on surface tension and hygroscopicity in lines 352-362,

"This effect was closely related to the presence of liquid-liquid phase separation (LLPS) (Renbaum-Wolff et al., 2016), which was observed in organic-containing particles under high relative humidity. LLPS is mainly depended on the chemical composition of organics (e.g., functional groups and oxidation state) and inorganic-organic mixing ratio (Ruehl et al., 2016; Ma et al., 2021; Bertram et al., 2011). Once LLPS occurred, organic-rich phase on the droplet surface would reduce surface tension and further enhance water uptake (Rastak et al., 2017;Freedman, 2017). Surface tension is expected to increase with droplet growth, since the organic-rich phase becomes thinner and shifted to water-rich phase (Liu et al., 2018; Renbaum-Wolff et al., 2016; Ovadnevaite et al., 2017). Further laboratory and field studies are needed for better understanding the occurrence of LLPS in particles, its variation with different chemical composition, and its impact on the surface tension."

**Minor comments:**

1.  A schematic diagram of the experimental setup is recommended.

Reply: We added a schematic diagram in section 2.1 and rephrased some sentences in lines 144-146:

"Two aerosol sampling ports equipped respectively with a $PM_{10}$ impactor and a $PM_{2.5}$ impactor were made of a 6 m long 3/8″ o.d. stainless-steel tube. The schematic diagram of the inlet system and instrument setup is shown in Fig. S1."

[Figure]

Figure S1. Schematic diagram of the experimental setup

2. The lowest measurable particle diameter in this study is 1 nm. Is there any reason not to use this for the estimation of formation and growth rates, instead of 3 nm (lines 227, 253, 265)?

Reply: The particle number size distribution (PNSD) data during the campaign was acquired by a commercial Nano-SMPS instrument. The instrument is controlled by Aerosol Instrument Manager (version 10, TSI Inc., USA) which does not provide accurate corrections for multiple charges and diffusion losses for particles smaller than 3 nm. While accurate inversion for particles smaller than 3 nm is still under development, we believe that it is adequate to use particles larger than 3 nm for modeling NPF in this study. Hence, we only used PNSD for particles larger than 3 nm to calculate formation and growth rates in this study. We added several sentences to clarify this issue in lines 159-163,
"The data inversion processes for the measured PNSD were done by Aerosol Instrument Manager (version 10, TSI Inc., USA). However, accurate inversion for particles smaller than 3 nm is currently still lacking due to large uncertainties from corrections for multiple charges and diffusion losses. Thus, we only discussed PNSD for particles larger than 3 nm in this study."

3. Line 415 and other instances, the "fail" is misspelled as "fell."
   Reply: Typos have been corrected in lines 436, 443 and 446.

**Reference:**

[revised manuscript text omitted]

---

## Author Comment (AC2) · 12 Mar 2021

We would like to appreciate the reviewer for providing valuable comments on our manuscript, and we have carefully addressed these comments point-by-point as follows. Please find the response (in red) to each comment below.

**Referee comments:**

Cai et al. present measurements of how new particle formation events and hygroscopicity impact cloud condensation nuclei concentrations. These observations were done in Guangdong Province at a rural site. The authors also modeled particle formation to see how it would impact CCN concentrations. The manuscript was at times difficult to understand due to frequent typos and unstructured paragraphs. However, the science done by the authors is sound as it has been implemented by numerous groups before. Also, their conclusions are logical. This paper fifits ACP and should be published once the authors address the below comments.

**Major comments:**

1.  The authors used a DEG SMPS to measure the particle size number distribution but instead used formation rate at 3 nm and growth rate from 3-60 nm. Is there a reason the <3 nm bins were not used? I would think the growth rate from 1-3 nm would play an even larger role in their particle number concentration modeling done in section 3.3 Can the authors comment on this and maybe look into their 1-3 nm data to see how that would impact their data analysis?

Reply: The particle number size distribution (PNSD) data during the campaign was acquired by a commercial Nano-SMPS instrument. The instrument is controlled by Aerosol Instrument Manager (version 10, TSI Inc., USA) which does not provide accurate corrections for multiple charges and diffusion losses for particles smaller than 3 nm. While accurate inversion for particles smaller than 3 nm is still under development, we believe that it is adequate to use particles larger than 3 nm for modeling NPF in this study. Hence, we only used PNSD for particles larger than 3 nm to calculate formation and growth rates in this study. We added several sentences to clarify this issue in lines 159-163,

"The data inversion processes for the measured PNSD were done by Aerosol Instrument Manager (version 10, TSI Inc., USA). However, accurate inversion for particles smaller than 3 nm is currently still lacking due to large uncertainties from corrections for multiple charges and diffusion losses. Thus, we only discussed PNSD for particles larger than 3 nm in this study."

2.  Page 10, line 207: The authors did not directly measure sulfuric acid concentration but instead used a proxy based on SO2. How accurate is the proxy for the region they are measuring in? Every 1-2 years, a new proxy sulfuric acid paper is published from a new measurement location. Each of these proxy equations is different, with different parameters and different coefficients. See (Mikkonen et al., 2011). Why did the authors choose this proxy? Could they provide some gauge of uncertainty as a function of time? The authors say 40% (line 313) but how was this estimated and how does it vary with CS and OH concentrations? This would be especially useful as the authors compare trends of when sulfuric acid concentrations peak at specific times of the day compared to growth rate and when hygroscopicity increased (page 19 line 397).

Reply: We agree with the reviewer that many different proxies have been proposed with different parameterizations for the estimation of sulfuric acid concentration. We adopted a proxy proposed by Lu et al. (2019) to calculate the $H_2SO_4$ concentration as shown below,

$$[H_2SO_4] = 0.0013 \cdot UVB^{0.13} \cdot [SO_2]^{0.40} \cdot CS^{-0.17} \cdot ([O_3]^{0.44} + [NO_x]^{0.41}) \tag{1}$$

This proxy was derived based on measurements from a winter field campaign in urban Beijing, where the atmospheric environment was similar to the locations of our measurements, which can provide a reasonable estimation for the $H_2SO_4$ concentration. Although accurate quantification of the uncertainty of the proxy is not feasible, we adopted a relative error of about 20% proposed by Lu et al. (2019) when applying the proxy for the estimation of sulfuric acid concentration. While, the relative error could be underestimated, since we ignored the uncertainty in measuring UVB, $[SO_2]$, CS, $[O_3]$ and $[NO_x]$. The relationship between uncertainty and CS and OH concentration was difficult to estimate, because we do not have a direct measurement of $[H_2SO_4]$ in this campaign. To be more clarified, we have made several modifications in lines 208-213,

"The daytime gas phase $H_2SO_4$ concentration is estimated according to the proxy proposed by Lu et al. (2019),

$$[H_2SO_4] = 0.0013 \cdot UVB^{0.13} \cdot [SO_2]^{0.40} \cdot CS^{-0.17} \cdot ([O_3]^{0.44} + [NO_x]^{0.41}) \tag{5}$$

where UVB is the ultraviolet radiation B intensity (in W m$^{-2}$), $[SO_2]$ is the concentration of $SO_2$ in molecules cm$^{-3}$, $[O_3]$ is the concentration of $O_3$ in molecules cm$^{-3}$, $[NO_x]$ is the concentration of $NO_X$ in molecules cm$^{-3}$, and the CS is the condensation sink and it can be calculated from following equation…"

and lines 218-225,

"This proxy is derived based on measurements from a winter field measurement in urban Beijing, where the atmospheric environment is similar to the locations of our measurements. A relative error of about 20% for the proxy proposed by Lu et al. (2019) was estimated based on comparison of the estimated $[H_2SO_4]$ with the measured one. However, accurate quantification of the uncertainty is not feasible since this proxy has not been tested in the PRD region. For simplicity, we adopted a relative error of about 20% for the estimation of sulfuric acid concentration, and ignoring the uncertainties in measuring UVB, $[SO_2]$, CS, $[O_3]$ and $[NO_X]$. However, further investigation is still needed, since the relative error of the estimation could vary temporally and spatially (Mikkonen et al., 2011)."

and lines 318-326:

"The average calculated $H_2SO_4$ concentration during particle formation periods (10:00-12:00 LT) was about $1.4 \times 10^7$ cm$^{-3}$, about an order higher than that (about 7 - $12 \times 10^6$ cm$^{-3}$) in a mountain region in Germany (Wu et al., 2013a) and close to that (about $2$-$5 \times 10^7$ cm$^{-3}$) in a rural region of Sichuan in China (Chen et al., 2014). Considering a relative error of about 20%, the growth rate contributed by condensation of gaseous $H_2SO_4$ was about 0.61-1.09 nm h$^{-1}$, or about 7.6% -13.6% of the observed growth rates for 3-10 nm particles. It should be pointed out that the above estimates for the growth rates are subject to large uncertainties due to uncertainties for the estimation of sulfuric acid concentration using Eq. (5) as the proxy and here a unity of sticking coefficients was assumed when gaseous $H_2SO_4$ molecules collide with pre-existing particles."

We have also modified Figure 1.

[Figure]

Figure 1. The PNSD (a), $N_{CN}$, $N_{CCN}$ and AR (b), wind speed and wind direction (c), $j_{O(1D)}$, and concentration of calculated $H_2SO_4$ (d) during the NPF event on 29th October, 2019. The blue dots in (a) represent the geometric mean particle diameter ($Dp_{gmd}$) and the red line represents the linear fitting.

3. Generally, the terminology used is confusing. Particle formation rate is used to described nucleation and of CCN (page 19 line 389). The convention is newly formed particles are small (<10 nm). It would be helpful if the authors could find a better phrase to call 30-40 nm particles. Also, measured, averaged, recalculated D50 are used. However, in the text, they often just say D50. Which D50 is it? Example (though not the only one) is page 19 line 403, page 20 line 406, etc. Maybe add a subscript to the D50 to make it clearer? Page 21 line 436: How does doubling the GR result in increasing particle concentration because of higher coagulation source? More coagulation would mean lower particle concentrations?

Reply:

(1) In order to avoid confusion, these particles were referred to as "newly-grown particles", since they were grown from newly-formed particles. We have modified corresponding

sentences in lines 411-414, "The hygroscopicity of newly-grown particles can have significant impact on the $N_{CCN}$ during the NPF event. During the campaign, the minimum particle size of CCN activity measurement was about 40-45 nm (at 1.0% SS), thus the hygroscopicity of this size range was used to present the property of the newly-grown particles, when they grow up to this size range.", and lines 418-423, "It should be pointed out that the high κ values during 10:00~12:00 LT did not represent the hygroscopicity of the newly-grown particles which were primarily composed of particles much smaller than 30-40 nm. Those new particles grew to about 40-50 nm at 14:00-16:00 (Fig. 1a and Fig. 3) and their κ values were obviously lower than the average ones, implying that the organic vapors could play an important role during growth of new particles as discussed in Section 3.1.", and lines 430-431, "As discussed in section 2.3.4, the dynamical processes for new particles during nucleation events are governed by the population balance equation (Eq. (13))."

(2) To be clarified, the measured $D_{50}$, recalculated $D_{50}$ and average $D_{50}$ were denoted as $D_{50,m}$, $D_{50,r}$ and $D_{50,a}$, respectively. We modified the corresponding sentences in section 3.2, Figure 3 and Figure 4.

(3) Coagulation means that two smaller particles collide with each other and become a larger particle, which can also increase the population of new particles. It is the third term on the right-hand side of Eq. (13-2),

$$\frac{1}{2}\sum_{Dp_i=Dp_{min}}^{k-1}\beta_{(i,\varphi)}N_iN_\varphi \tag{13-2}$$

A higher GR would lead to a wider distribution of new particles, since these particles can grow to a larger size in the same time. This thus provides a wider "region" for coagulation sources, i.e., the "$k-1$" in Eq. (13-2) is higher. In order to avoid any confusion, we modified the sentences in lines 460-462, "Coagulation source means that two smaller particles collide with each other and become a larger particle, which can also increase the population of new particles.", and lines 464, ", i.e. the "$k-1$" in eq. (13-2) is higher".

4. Page 21 paragraphs starting with line 427: This section is very repetitive and difficult to understand. What do the authors want us to take away from this section? Is there a more efficient way of communicating this information then just listing off every possible configuration of the model? The findings in this section are not new science so sticking to highlights of why this section is needed to convey the purpose of the paper would be helpful.

Reply: We thank the review for this valuable comment. The contribution of NPF to the CCN is not only affected by the properties of newly-grown particles, but also affected by characteristics of NPF, including formation rate, growth rate and background particle number size distribution (PNSD). However, there is still lack of understanding on the major impact factors among these parameters and how these factors compare with particle properties (e.g., hygroscopicity). The two paragraphs (lines 427-475 on the original version) are included to investigate the most important factor that contributes to the CCN concentration. We found that high growth rate significantly affected the variation of $N_{CCN}$, and high background particle concentration could hinder growth of new particles to the CCN sizes. For better clarification, we added several sentences to begin the paragraphs in

lines 448-450, "As discussed in section 3.2, the contribution of $N_{CCN}$ was affected by the properties of newly-grown particles. However, the characteristics of NPF, including growth rate, formation rate and the background PNSD also affect $N_{CCN}$.", and a comparison with particle properties was included in lines 501-513, "To compare different impacts of the characteristics and properties of newly-formed particles, the $N_{CCN}$ was simulated through varying parameters of different characteristics (case 1, 4 and 7) and properties (case 2, 3, 5, 6, 8 and 9). The input parameters for different cases are shown in Table S1. For case 2, 3, 5, 6, 8 and 9 scenarios, the surface tension or hygroscopicity was adjusted to match similar $N_{CCN}$ values based on different NPF characteristics (case 1, 4 and 7, respectively). The results show that doubling GR produces the most significant impact on the $N_{CCN}$, and the surface tension ($\kappa$ value) was adjusted to 0.030 N $m^{-1}$(1.2) to have the same impact (Fig. 8a). Obviously, a $\kappa$ value of 1.2 for hygroscopicity is much higher than that of many inorganics, e.g., $H_2SO_4$ ($\kappa$=0.90, Topping et al., 2005) and $NH_4NO_3$ (0.58, Topping et al., 2005). Meanwhile, the surface tension was lower than the values (0.049-0.060) reported previously (Ovadnevaite et al., 2017; Engelhart et al., 2008; Cai et al., 2018). However, doubling GR value (16.0 nm $h^{-1}$) was reasonable and consistent with previous studies (Mönkkönen et al., 2005; Foucart et al., 2018; O'Dowd et al., 1999), suggesting significant contribution of GR to the growth. For doubling formation rate and halving PNSD, the modified surface tension and $\kappa$ values were minor (Fig. 8b and c)."

Table S1. The input parameters for Case 1-9

$2\times$ and $0.5\times$ represent doubling and halving the parameters, respectively.

|  | GR | J | PNSD | $\sigma_{s/a}$ (N m$^{-1}$) | $\kappa$ |
|---|---|---|---|---|---|
| Case 1 | $2\times$ | $1\times$ | $1\times$ | 0.0728 | Measured |
| Case 2 | $1\times$ | $1\times$ | $1\times$ | 0.030 | Measured |
| Case 3 | $1\times$ | $1\times$ | $1\times$ | 0.0728 | 1.2 |
| Case 4 | $1\times$ | $2\times$ | $1\times$ | 0.0728 | Measured |
| Case 5 | $1\times$ | $1\times$ | $1\times$ | 0.065 | Measured |
| Case 6 | $1\times$ | $1\times$ | $1\times$ | 0.0728 | 0.15 |
| Case 7 | $1\times$ | $1\times$ | $0.5\times$ | 0.0728 | Measured |
| Case 8 | $1\times$ | $1\times$ | $1\times$ | 0.067 | Measured |
| Case 9 | $1\times$ | $1\times$ | $1\times$ | 0.0728 | 0.13 |

[Figure]

Figure. 8 The model $N_{CCN}$ based on different characteristics (doubling growth rate and formation rate, and halving background particle distribution) and particle properties. Different colors and markers represent case 1-9, respectively.

”

**Minor comments:**

1. Line 27: environments and depend.

Reply: It has been revised.

2. Line 29: investigates.
Reply: It has been revised.

3. Line 38: than assuming pure water.
Reply: It has been revised.

4. Line 43 fact.
Reply: It has been revised.

5. Line 51: controlling factor is a weird phrase. What do you mean by controlling?
Reply: We had meant factors (e.g., growth rate, formation rate, hygroscopicity) which play important roles in affecting the CCN activity. To avoid confusion, it has been revised to "the major impact factors".

6. Line 56-60 these two sentences are wordy and difficult to understand.
Reply: They have been revised to "In general, atmospheric particles have a cooling effect on the global climate with the highest uncertainty among all the climatic forcings (Stocker et al., 2013). The relationship between the CCN number concentration ($N_{CCN}$) and its climatic effect represents one of the major uncertainties."

7. Line 61: controlling factors, again what do you mean by this?
Reply: It has been revised to "the major impact factors".

8. Line 63: also marine?
Reply: "marine" has been added in this sentence.

9. Line 75: matter
Reply: It has been revised.

10. Line 76: and more easily
Reply: It has been revised.

11. Line 77: matter
Reply: It has been revised.

12. Line 80: awkward statement of constraining an accurate quantification of the aerosol…
Reply: It has been modified to "…which becomes a challenging in quantification of the climatic forcing of NPF events."

12. Line 82: NPF event is
Reply: It has been revised.

13. Line 103: growth rates

Reply: It has been revised.

14. Line 124: have reported

Reply: It has been revised.

15. Line 135: that manipulate is awkward phrase.

Reply: It has been revised to "affect".

16. Line 234: necessarily the case, also this entire sentence should be rewritten to be more clear.

Reply: It has been revised to "…not necessarily the case because not all $H_2SO_4$ molecules will be captured when colliding with the particles."

17. Line 265: represents the smallest detectable particle size. The smallest stable size is likely much smaller than 3 nm.

Reply: It has been revised.

18. Line 357: D50 is shown.

Reply: It has been revised.

19. Line 359: shows a difference of what?

Reply: It has been revised to "shows a difference between the $D_{50,r}$ and the $D_{50,m}$."

20. Page 20: all fell should be failed

Reply: It has been revised.

21. Line 429: measured NCN and NCCN and the modeled one is awkward phrasing.

Reply: It has been modified to "… the comparison of the measured $N_{CN}$ and $N_{CCN}$ with their respective modeled values…."

22. Line 451: what is double background PNSD condition?

Reply: "Double background PNSD condition" means the background PNSD as an input parameter is doubled compared to the standard characteristic. It is clarified at the beginning of this paragraph in lines 450-452.

23. Line 454: NPF events a minor contribution, awkward phrasing

Reply: It has been revised to "NPF events have a minor contribution".

24. Line 547: profound impact, awkward phrasing

Reply: It has been revised to "significant impact".

25. Line 758: linear fit

Reply: It has been revised.

26. Line 772: space after activation ratio

Reply: It has been revised.

27. Figure 2: is there a way to not use shades of the same color on this graph? The blues
cannot be easily distinguished. Maybe adding symbols would help?

Reply: We changed the color of "κ$_{HTDMA}$ (This Measurement)" from light blue to black.

[Figure]

Figure 2. The median and interquartile κ obtained from HTDMA and CCN measurements during
this campaign, at the Panyu site (urban Guangzhou), and from South China Sea. The κ was pointed
against the corresponding median $D_{50}$ (CCN measurement) or selected diameter (HTDMA
measurement). Dots represent the median values and bars represent the interquartile ranges. The κ
values in this measurement were obtained from HTDMA measurement (in black) and CCNc
measurement (ss=0.1%, 0.2%, 0.4%, 0.7%, 0.9%, and 1.0% in red and yellow for different surface
tensions). The yellow lines and dots represent the κ values recalculated based on $\sigma_{s/a}^{*}$. The κ values
in the Panyu measurement were obtained from HTDMA measurement (in purple) and CCNc
measurement (ss=0.1%, 0.2%, 0.4%, and 0.7%, in green). The κ values from the South China Sea
were obtained from CCNc measurement (ss=0.18%, 0.34%, and 0.59%, in light blue). The κ values
from the North China Plain were obtained from HTDMA measurement.

28. Figure 6: maybe helpful to write in the legend 0.5xGR and 2xGR, etc. to make it clearer.

Reply: It has been revised. We also revised Fig. 6, Fig. S5, Fig. S6, and Fig. S7.

29. Table S1: background particle distributions

Reply: It has been revised.

30. Figure S2: Is average value during the campaign mean it was averaged over that time of day
interval over the whole campaign?

Reply: Yes. It represents the average diurnal variation during the whole campaign. We had revised
the title to make it clear, "…The blue color represents the average diurnal variation during the
campaign.".

31. Figure S7: What is it meant by "new" GR, formation rate, and background PSND?

  Reply: They represent the GR on the October 18 event (highest among three events), formation rate on the October 29 event (highest among three events) and background PNSD (lowest CS among three events) on the October 29 event. To avoid confusion, we have revised the corresponding sentences in lines 552-555 and Fig. 11 and Fig. S9, "…including the growth rate on the October 18 event (high growth rate scenario), the formation rate on the October 29 event (high formation rate scenario), and the background PNSD on the October 29 event (mainly distributed in Aitken mode, denoted as low CS PNSD scenario…"

[Figure]

Figure 11. The measured and model $N_{CN}$ (a) and $N_{CCN}$ (b) on the Panyu NPF event. The bule line represents the measured value. The red, yellow, purple and green lines represent the simulated $N_{CCN}$ based on standard input, growth rate of the NPF event on October 18[th] (denoted as high GR), formation rate of the NPF event on October 29[th] (high J), and background particle distribution of the NPF event on October 29[th] (low CS PNSD), respectively.

[Figure]

Figure S9. The simulated NPF event on 12th December, 2014 based on the high growth rate (a), the high formation rate (b), and the low CS PNSD (c).

**Reference:**

Lu, Y., Yan, C., Fu, Y., Chen, Y., Liu, Y., Yang, G., Wang, Y., Bianchi, F., Chu, B., Zhou, Y., Yin, R., Baalbaki, R., Garmash, O., Deng, C., Wang, W., Liu, Y., Petäjä, T., Kerminen, V. M., Jiang, J., Kulmala, M., and Wang, L.: A proxy for atmospheric daytime gaseous sulfuric acid concentration in urban Beijing, Atmos. Chem. Phys., 19, 1971-1983, 10.5194/acp-19-1971-2019, 2019.

Mikkonen, S., Romakkaniemi, S., Smith, J., Korhonen, H., Petäjä, T., Plass-Duelmer, C., Boy, M., McMurry, P., Lehtinen, K., and Joutsensaari, J.: A statistical proxy for sulphuric acid concentration, Atmos. Chem.    Phys., 11, 11319-11334, 2011.